# Total ozone column from OMPS-NM measurements using the broadband Weighting Function Fitting Approach (WFFA)

Andrea Orfanoz-Cheuquelaf[1], Alexei Rozanov[1], Mark Weber[1], Carlo Arosio[1],
Annette Ladstätter-Weißenmayer[1], and John P. Burrows[1]

[1]Institute of Environmental Physics, University of Bremen, Otto-Hahn-Allee 1, D-28359 Bremen, Germany

**Correspondence:** Andrea Orfanoz-Cheuquelaf (andrea@iup.physik.uni-bremen.de)

**Abstract.** A scientific total ozone column product from the Ozone Mapping and Profiler Suite Nadir Mapper (OMPS-NM) observations and the retrieval algorithm are presented. The retrieval employs the Weighting Function Fitting Approach (WFFA), a modification of the Weighting Function Differential Optical Absorption Spectroscopy (WFDOAS) technique. The total ozone columns retrieved with WFFA are in very good agreement with other datasets. A mean difference of 0.3 % with respect to ground-based Brewer and Dobson measurements is observed. Seasonal and latitudinal variations are well represented and in agreement with other satellite datasets. The comparison of our product with the operational product of OMPS-NM indicate a mean bias of around zero. The comparison with the Tropospheric Monitoring Instrument products (S5P/TROPOMI) OFFL and WFDOAS, shows a persistent negative bias of about -0.6 % for OFFL and -2.5 % for WFDOAS. Larger differences are only observed in the polar regions. This data product is intended to be used for trend analysis and the retrieval of tropospheric ozone combined with the OMPS limb profiler data.

## 1 Introduction

The majority of the ozone's atmospheric load ($O_3$) resides in the stratosphere. The strong absorption of the Ultraviolet (UV) B and C radiation by $O_3$ shields the biosphere from biologically damaging UV radiation. $O_3$ heats the atmosphere and creates the temperature inversion. This plays a key role in determining the tropopause height and influences tropospheric weather. Anthropogenic emissions lead to its production in the lower atmosphere. Exposure to this secondary air pollutant causes health problems and vegetation damage (e.g., Schultz et al., 2015; Mills et al., 2018). As tropospheric ozone is a potent greenhouse gas and an essential climate variable, knowledge about the global amount and evolution of this gas is needed, which can only be provided by satellite measurements. Global ozone distribution can be derived using nadir satellite observations.

Since the 1970's, satellite instruments have provided a global picture of total ozone amounts using nadir viewing geometry. The Backscatter Ultraviolet Ozone experiment (BUV, 1970-1976), superseded by the Solar Backscatter UltraViolet (SBUV, 1978-1990) and the SBUV/2 instrument series (since 1985), the Total Ozone Mapping Spectrometer (TOMS, 1978-2005), the Ozone Monitoring Instrument (OMI, 2004-present) and the Ozone Mapping and Profiler Suite (Suomi NPP OMPS, 2011-present), provide total ozone column (TOC) products, sharing the same operational retrieval approaches, known as TOMS (all instruments) and SBUV algorithms (SBUV only) (Labow et al., 2013; Bramstedt et al., 2003; McPeters et al., 2015; Flynn et al.,

2004; Bhartia, 2002). The Global Ozone Monitoring Experiment (GOME, 1995-2011) (Burrows et al., 1999), the SCanning Imaging Absorption spectroMeter for Atmospheric CHartographY (SCIAMACHY, 2002-2012) (Bovensmann et al., 1999) and GOME-2 (2006-present) (Munro et al., 2016) also provide TOC products using the differential optical absorption spectroscopy (DOAS) approach (Hao et al., 2014; Van Roozendael et al., 2006).

The measurements of total ozone have also been used in the determination of the tropospheric ozone amount. A widely
used approach for that is the residual technique (Fishman and Larsen, 1987). With this technique, the tropospheric ozone is determined by subtracting the stratospheric column retrieved from limb observations from the total ozone column retrieved from another instrument's nadir observations. This was indeed one motivation to build the pioneering SCIAMACHY instrument, which performed alternating measurements in the nadir and limb viewing geometries from 2002 to 2012 (Burrows et al., 1995). Ebojie et al. (2014) combined for the first time nadir and limb observations from the same instrument, SCIAMACHY.
OMPS features a combination of limb (LP) and nadir sensors (NM), similar to SCIAMACHY. To use OMPS data to retrieve tropospheric $O_3$ with the limb-nadir matching technique and generate a consistent long term dataset by combining OMPS data with SCIAMACHY, we developed a scientific TOC product from OMPS-NM observations.

The retrieval approach adapts the Weighting Function-DOAS technique (WFDOAS), successfully applied for SCIAMACHY (Bracher et al., 2005), GOME (Weber et al., 2005) and GOME-2 (Weber et al., 2007), for the use with OMPS-NM measure-
ments and is referred to as Weighting Function Fitting Approach (WFFA). While the DOAS technique relies on the retrieval from differential absorption only, the WFFA technique uses both the differential structure and the broadband spectral signature of the ozone absorption in the UV spectral range. The latter works better for instruments with a coarser spectral resolution than GOME or SCIAMACHY, such as OMPS.

The WFFA total ozone retrieval has been specifically developed for combining it with the limb ozone profile retrieval from
OMPS-LP to retrieve tropospheric $O_3$ and continue with the heritage of SCIAMACHY.

The OMPS-NM instrument and the input data used are described in Section 2. A description of a new a priori ozone profile climatology used in the retrieval is given in Section 3. The WFFA retrieval algorithm is presented in Section 4. Section 5 introduces the datasets used for the validation, and the validation results of the OMPS-WFFA TOC are presented in Section 6.

## 2  OMPS-NM

The Ozone Mapping and Profiler Suite (OMPS) is one of the five instruments on board the Suomi National Polar-orbiting Partnership (Suomi NPP). This satellite is part of the Joint Polar Satellite System Program (JPSS), a collaborative program between the National Oceanic and Atmospheric Administration (NOAA) and the National Aeronautics and Space Administration (NASA) (Goldberg and Zhou, 2017). Suomi NPP was launched on October 28th, 2011, has a sun-synchronous orbit with 13:30 ascending node, flies at a mean altitude of 824 km and performs fourteen orbits per day.
OMPS is a three-part instrument, namely a nadir mapper (OMPS-NM), a nadir profiler (OMPS-NP) and a limb profiler (OMPS-LP) sensor, collecting data since January 2012. OMPS-NM was designed to accomplish total column retrieval, using a two-dimensional charge-coupled device (CCD). The spectrometer registers backscatter solar radiation every 0.42 nm between

300 to 380 nm, with a spectral resolution of 1 nm. The footprint of OMPS-NM is approximately 50 x 2800 $km^2$, with 0.27° along-track field of view (FOV) and 110° across-track FOV divided into 36 bins. The two central FOVs cover 50 km x 20 km and 50 km x 30 km, the rest, approximately 50 km x 50 km each (Flynn et al., 2004, 2014; Seftor et al., 2014).

For the retrieval of OMPS TOC, the level 1 data, version 2.0 (L1b V2.0), of OMPS-NM were used (Jaross, 2017a). So far, the limb ozone profiles are only retrieved from the central slit of the three vertical slits of OMPS-LP (Arosio et al., 2018), resulting in a horizontal sampling of about 150 km along-track and 3 km across-track (Rault et al., last access: 23 June 2021). In order to match our nadir TOC product to OMPS limb profiles for obtaining tropospheric ozone columns, only the central OMPS-NM across-track FOV bins, 10 to 22, are needed and were processed (approximately 50 km x 600 km wide swath). Only pixels with cloud fractions under 0.1 and solar zenith angles smaller than 80° were used. The period for that the ozone data are to be retrieved is intended to cover the years from 2012 until 2018. Currently, only the data from 2016 to 2018 has been retrieved. Later data were not considered because of systematic errors in measured radiances of OMPS-LP (Kramarova et al., 2018) that lead to a significant drift in OMPS-LP ozone, which would affect the tropospheric ozone. The cloud fraction and topography information from OMPS-NM Level 2 (L2) version 2.1 product was used as input in the retrieval.

## 3  A priori ozone profile climatology

It is well known that a good knowledge of the ozone profile shape helps to increase the quality of TOC retrievals from nadir measurements in the UV spectral range. As discussed by Lamsal et al. (2007), differences in the retrieved total ozone due to a priori ozone profile might go up to 10 %. Most of the ozone climatologies available so far were created from periods before the year 2012 (McPeters et al., 1997; Paul et al., 1998; Lamsal, 2004; McPeters et al., 2007; Labow et al., 2015; Yang and Liu, 2019). Therefore, it was decided to create a new ozone profile database to have a consistent input for the time frame of this retrieval, by using OMPS-LP (Arosio et al., 2018) and ozonesonde observations between January 2012 and December 2018.

The ozone profiles are provided as a function of latitude band, season, and total ozone content as in the ozone climatology from Lamsal (2004). Therefore, the ozone database consists of zonally and latitudinally averaged profiles for five regions: northern polar region (np, 60°-90° N), northern mid-latitudes (nm, 30°-60° N), tropics (trop, 30°N-30° S), southern mid-latitudes (sm, 30°-60° S), and southern polar region (sp, 60°-90° S). Due to the typical annual cycle of the total ozone column, the profiles have been classified in two groups considering the season: winter/spring (ws) and summer/fall (sf), except for the tropics, where no seasonality was considered. The final profiles were grouped and averaged by their total ozone column amount in intervals of 30 DU. For each ozone profile, a temperature profile is provided as well but is not used in the retrieval.

As the total ozone retrieval is sensitive to changes in the ozone profiles in both the stratosphere and the troposphere (Wellemeyer et al., 1997), the database was built by combining stratospheric profiles from OMPS-LP and ozonesonde measurements for the troposphere. The limb profiles are from the scientific zonal average Level 3 product from OMPS-LP provided by Arosio et al. (2018), that contains gridded monthly means between January 2012 and December 2018. These profiles are zonal averages, every 5° in latitude, for 53 altitudes from 8.5 to 60.5 km with a sampling of 1 km. Here, the profiles from 12.5 km altitude up to the top-of-the atmosphere were used. The ozonesondes data used are from the World Ozone and Ultraviolet Data

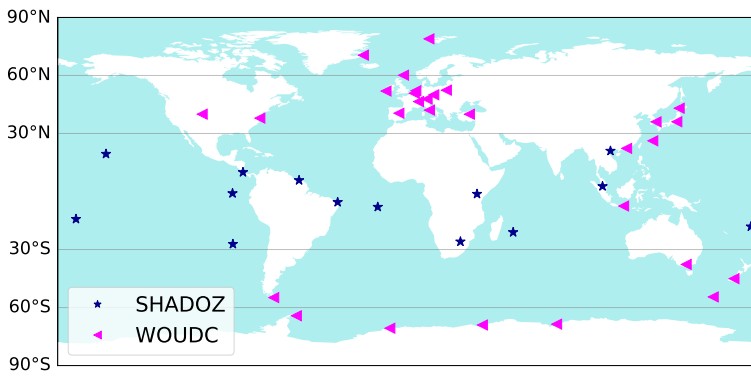

**Figure 1.** Map of the ozonesonde launch sites included in ozone profiles database. Blue stars are the stations from SHADOZ (14 in total) and pink triangles the stations from WOUDC (29 stations). The horizontal lines mark the zonal bands used in the classification of the new ozone climatology.

Center (WOUDC) (Fioletov et al., 1999) and from the Southern Hemisphere Additional Ozonesondes (SHADOZ) (Thompson et al., 2007). All stations with data between 2012 and 2018 were used, 29 stations from WOUDC and 14 from SHADOZ (Fig. 1). Each ozonesonde profile was convolved using a Gaussian function with 3.3 km FWHM, to obtain a resolution similar to that of the OMPS-LP profiles (Arosio et al., 2018), and sampled onto a grid of 1 km from 0.5 to 20.5 km.

Every ozone profile in the database was created using the ozonesonde profile up to 11.5 km and the zonal monthly mean limb profile above 20.5 km. In the transition zone between 12.5 and 20.5 km, the merged profile results from a linearly weighted average between the ozonesonde and the limb profile. Each ozonesonde profile was joined with the corresponding zonal monthly mean stratospheric profile, matching the latitude and the month of the ozonesonde. These merged profiles were averaged considering their total ozone content, date, and latitude according to the description above. The resulting ozone climatology profiles in volume mixing ratio units are shown in Fig. 2.

## 4   Retrieval algorithm

The retrieval algorithm used here is a modification of the Weighting Function Differential Optical Absorption Spectroscopy algorithm (WFDOAS) which has been developed for the retrieval of trace gases in the near-infrared spectrum range from SCIAMACHY measurements (Buchwitz et al., 2000). It was adapted and successfully applied for TOC retrieval in the UV spectral range, from nadir viewing measurements of GOME (Coldewey-Egbers et al., 2005), GOME2 and SCIAMACHY (Weber et al., 2005; Bracher et al., 2005; Weber et al., 2007).

The algorithm approximates the measured atmospheric optical depth by a Taylor expansion around a first guess atmospheric state. Also, contributions from interfering species, not included in the forward model, and a polynomial are included in the fit (Coldewey-Egbers et al., 2005):

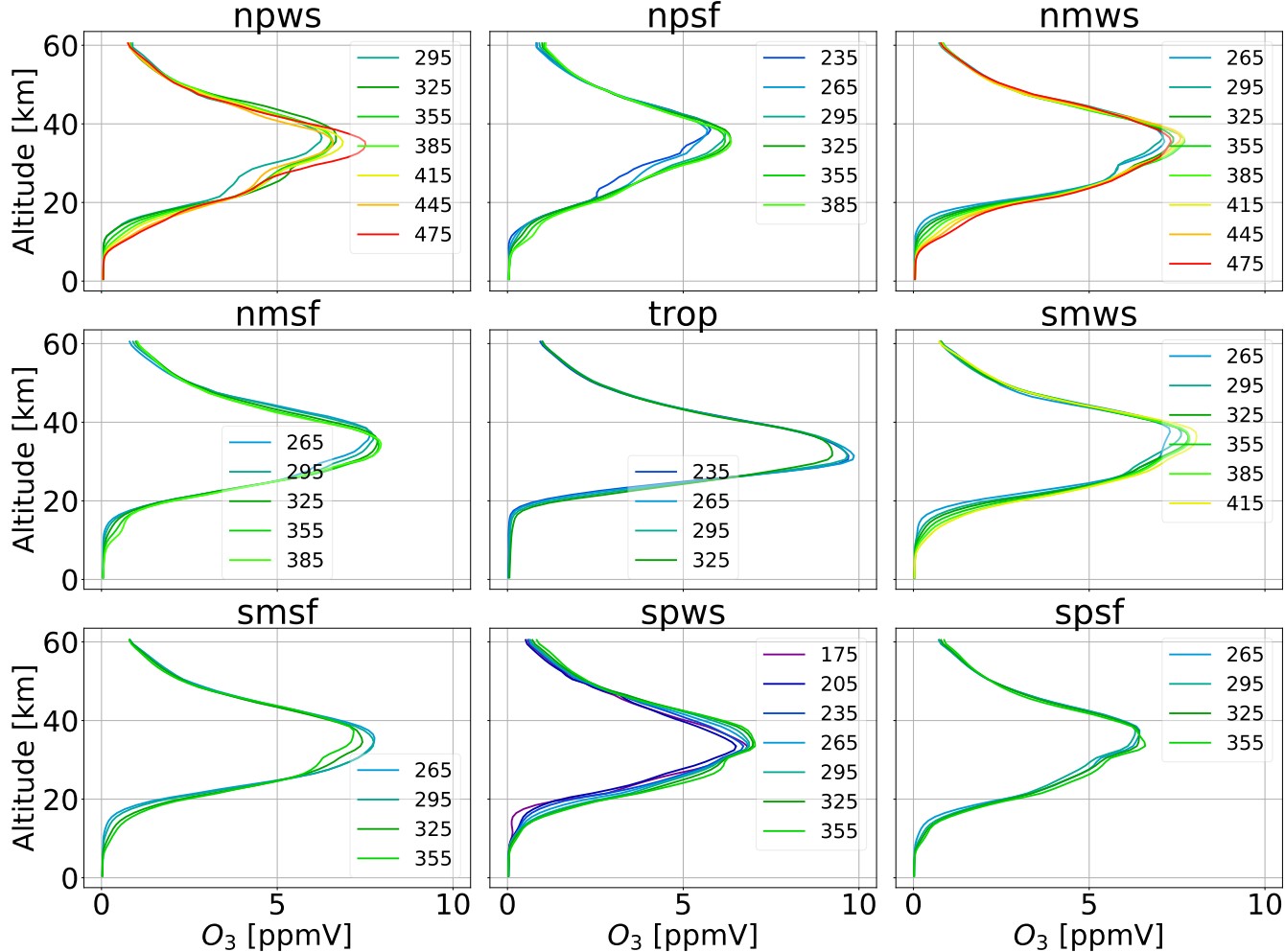

**Figure 2.** Profiles from the ozone a priori database, for each latitudinal region, season and total ozone classification. The labels indicate the total ozone concentration in DU. The titles indicate the region and season (see main text for details).

$$\ln I_i^{mea}(V^t, \mathbf{b}^t) \approx \ln I_i^{mod}(\overline{V}, \overline{\mathbf{b}}) + \frac{\partial \ln I_i^{mod}}{\partial V}\Big|_{\overline{V}} \times \Delta V + \frac{\partial \ln I_i^{mod}}{\partial T}\Big|_{\overline{T}} \times \Delta T \tag{1}$$
$$+ \quad SCD_{NO_2} \cdot \sigma_{i,NO_2} + SCD_{Ring} \cdot \sigma_{i,Ring} + C$$

For each ground pixel, the natural logarithm of the sun-normalized measured radiance ($I_i^{mea}$) is fitted by the natural logarithm of the modelled reference intensity ($I_i^{mod}$), the weighting functions of ozone ($\partial \ln I_i^{mod}/\partial V$) and temperature ($\partial \ln I_i^{mod}/\partial T$), the $NO_2$ cross-section $\sigma_{i,NO_2}$, as in the standard DOAS approach, the Ring spectrum $\sigma_{i,Ring}$, and a low order polynomial ($C$). In Eq. (1) the index $i$ references the wavelengths, $V^t$ is the true vertical ozone column, and $\mathbf{b}^t$ are true atmospheric conditions (pressure, temperature, albedo, etc.). $\overline{V}$ is the reference (i.e. used in the forward model) ozone column,

$\overline{T}$ is the reference temperature profile and $\overline{\mathbf{b}}$ is the atmospheric state as used in the forward model. $\Delta V$ and $\Delta T$ represent the corrections to the reference values as a result from the fit. The scalar correction to the temperature profile ($\Delta T$) is a shift applied to the entire vertical temperature profile.

When applying the standard WFDOAS approach to OMPS-NM measurements, the coarse spectral resolution of the latter was found to result in unstable retrievals. To adapt the retrieval technique, it was decided to use a lower order polynomial, a wider spectral window, and every second spectral point from the input radiance. In the WFDOAS approach, a cubic polynomial is usually used to account for all broadband contributions; consequently, the total column ozone information is obtained from the differential absorption structure only. For OMPS, this resulted in strong variations in the total ozone retrieved from different

ground pixels in the across-track direction (for details see Appendix A1). Therefore, a zero degree polynomial (a constant, $C$) is used instead of the cubic one and the broad-band spectral signature of ozone absorption is also fitted. To further reduce the impact of the differential ozone absorption structure in the fit, the spectral window was chosen to be 316-336 nm, which is wider than typically used in WFDOAS (325 to 335 nm). In addition, only the odd-numbered spectral points are used in the retrieval, counting from the first spectral point of the selected fitting window (see Appendix A2 for details). Even with a

wider spectral window, the use of either all spectral points or the even-numbered ones, in some cases, resulted in significant discrepancies in the retrieved TOC from ground pixel to ground pixel, and in a negative bias of around 2 % with respect to the preferred wavelength selection. The retrieval using the odd-numbered spectral points shows less dependence on the temperature in the fit as compared to other wavelength samples (Appendix A2). With these changes, we now refer to the retrieval method as the Weighting Function Fitting Approach, WFFA. Apart from using a low-order polynomial and the wider spectral fit

window, WFFA is similar to WFDOAS (Coldewey-Egbers et al., 2005). Some further modifications have been implemented, as described below.

    The fitting procedure follows an iterative scheme. First, the synthetic radiance and all weighting functions needed in Eq. (1) are computed with a radiative transfer model (RTM). To account for a possible wavelength misalignment between the earthshine spectrum and the solar reference spectrum, the wavelength grid of the earthshine spectrum is adjusted through an

iterative non-linear fit of the shift and squeeze parameters. In the second step, the fit parameters in Eq. (1) ($\hat{V}$, $\hat{T}$, $SCD_{NO_2}$, $SCD_{Ring}$) and the constant ($C$) are estimated using a linear least-squares minimization. The resulting total ozone is then passed to the RTM to start the next iteration. The iterative process is terminated when the retrieved ozone column differs by less than 1 DU from the result of the previous iteration.

    The reference intensities, as well as the weighting functions, are computed with the RTM SCIATRAN V4.2 (Rozanov et al.,

2014), using the ozone profile climatology described in Section 3, for a given total ozone, zonal band, and season. During the iterative procedure a new ozone profile is selected according to the retrieved total ozone amount. For each ground pixel, the pressure and temperature profiles are obtained from ECMWF ERA5 (Hersbach et al., 2020). For solar zenith angles (SZA) larger than 40° the pseudo-spherical approximation is employed, whereas for smaller SZAs the plane parallel atmosphere is used, which is faster. The pseudo-spherical approximation solves the radiative transfer equation for a plane parallel atmosphere,

however the single-scattering source function is calculated considering the spherical shape of the atmosphere. The ground level

viewing geometry is used in the forward model. Compared with the spherical mode (Rozanov et al., 2000), the use of this approach yields almost identical results (de Beek et al., 2004).

The selected initial guess value of total ozone for the first pixel processed per FOV is 300 DU. The subsequent pixels use as initial value the retrieved TOC from the previous one. The ozone absorption cross-sections from Serdyuchenko et al. (2014) and the $NO_2$ absorption cross-sections from Burrows et al. (1998) are used. An aerosol free atmosphere is assumed in the model. As in WFDOAS, the effective scene albedo is retrieved near 377 nm using the Lambert equivalent reflectivity (LER) approach (Coldewey-Egbers et al., 2005) (see Appendix A3 for estimation of the related uncertainties).

The Ring effect is estimated using the difference in the optical depths calculated by the SCIATRAN model with and without Raman scattering (Rozanov and Vountas, 2014). Lookup tables (LUT) of radiances accounting for the Ring effect, i.e. infilling of Fraunhofer lines and molecular absorption bands, were simulated using SCIATRAN V4.2 and implemented in the retrieval scheme. With the pixel's viewing geometry information, total ozone, surface albedo, and altitude, the LUT are read and interpolated to obtain the corresponding Ring spectrum at high spectral resolution. After convolution of the LUT radiances with and without Ring effect with the instrument response function, the logarithm of the ratio of both convolved radiances is used as the Ring spectrum in Eq. (1). A second lookup table provides modelled sun-normalized radiances calculated with and without polarisation. From these, correction factors are determined to convert the observed (polarised) radiances into scalar radiances. With the LUTs, the time-consuming RTM modelling of the Ring and polarisation effects during the retrieval can be avoided. As the Ring effect and polarisation depend on ozone, the inputs from the LUT are updated in each iteration.

A full analysis of uncertainties and errors was performed for WFDOAS and presented by Coldewey-Egbers et al. (2003). In addition, we checked the major sources of errors that could affect our retrieval differently due to the change of the fitting window. Table 1 presents the results of the sensitivity tests that include enhanced aerosol loading, choice of ozone absorption cross-section and tropospheric ozone profile shape. Details on the enhanced aerosols loading and the tropospheric ozone tests can be found in Appendixes A3 and A4, respectively.

## 5  Validation datasets

In order to evaluate our scientific product, a comparison with other total ozone column measurements was performed. The NASA product from OMPS-NM, the operational OFFL and scientific WFDOAS products from the Tropospheric Monitoring Instrument on board Copernicus Sentinel-5 Precursor (S5P/TROPOMI), and ground-based Brewer and Dobson measurements, were used.

### 5.1  Ground-based measurements

The comparison with ground-based data was performed using daily means of total ozone columns from 18 Dobson (Basher, 1982) and 30 Brewer (Kerr, 2002) stations, obtained from the WOUDC dataset. Only ozone data derived from direct sun (DS) measurements are included in the analysis as they are the most accurate (Vanicek et al., 2003).

**Table 1.** Main uncertainty sources of the WFFA technique.

| Error source | Percent error |
|---|---|
| Enhanced weakly absorbing boundary layer aerosols (large SZAs) | less than 0.5 % |
| Enhanced strongly absorbing boundary layer aerosols (large SZAs) | better than - 1 % |
| Extreme volcanic aerosol loading in the stratosphere (large SZAs) | $\approx 1$ % |
| Enhanced boundary layer aerosols | less than 3 % |
| Extreme volcanic aerosol loading in the stratosphere (small SZAs) | $\approx 8$ % |
| BDM (Malicet et al., 1995) vs Serdyuchenko cross-section | $< 1\%$ below 70° SZA<br>$< 2\%$ beyond 70° SZA |
| Tropospheric ozone profile shape | less than 0.01 % |

## 5.2 Operational OMPS-NM total ozone column

The operational OMPS-NM Level 2 (L2) version 2.1 total ozone column product (Jaross, 2017b) is generated using NASA's V8.5 total column retrieval algorithm. This algorithm uses a pair of wavelengths to retrieve cloud fraction and ozone, 317.5 and

331.2 nm for most conditions as well as 331.2 and 360 nm for high amounts of ozone and large SZAs (*OMPS Nadir Mapper Level 2 Description*). The weak ozone absorption wavelength (331.2 nm) is used to estimate effective surface reflectivity, and effective cloud fraction through the Mixed Lambert Equivalent Reflectivity model. The strong-absorbing wavelength (317.5 nm) is used to estimate ozone. The measured radiances are compared with a pre-calculated set of radiances using various ozone and temperature profiles, and the TOC is obtained using piece-wise linear interpolation (Bhartia, 2002).

The validation of the NASA data product was presented in McPeters et al. (2019). They performed comparisons with ground-based measurements, Dobson and Brewer stations, and with the Merged Ozone Data time series (MOD) (Frith et al., 2014), that, for the period of comparison with OMPS-NM, is a combination of SBUV/2 instruments on three different satellites, NOAA 16, 18, and 19. The comparison with ground-based instruments located in the northern hemisphere showed a very good agreement with differences to within 0.5 % and an average bias of less than 0.2 %, from April 2012 to the end of 2016.

Concerning MOD, monthly mean global average showed a bias of -0.2 %.

## 5.3 S5P/TROPOMI total ozone column

The Sentinel-5 Precursor (S5P) is the first of the atmospheric-composition Sentinel satellites, as part of the Copernicus Program. It was launched in October 2017, in a sun-synchronous orbit with 13:30 ascending node, approximately 5 minutes behind Suomi NPP carrying OMPS. The TROPOspheric Monitoring Instrument (TROPOMI) aboard S5P is a nadir viewing spectrom-

eter that provides measurements in the ultraviolet, visible, near-infrared and short wave infrared spectral bands. TROPOMI has a ground pixel resolution of 3.5 km x 7 km (3.5 km x 5.5 km since August 2019), covering 2600 km across-track (Veefkind et al., 2012).

The L2 product of S5P/TROPOMI used in this study is the offline (OFFL and RPRO) total ozone column product (Lerot et al., 2020). S5P/TROPOMI OFFL and RPRO total ozone are very similar and are obtained using the GODFIT version 4 retrieval (Lerot et al., 2014). The algorithm performs a direct comparison with simulated radiances through non-linear least-square inversion, using the sun-normalized measured radiance from 325 to 335 nm. The modelled radiances and Jacobians are obtained with the RTM LIDORT (Spurr et al., 2018).

A validation for S5P/TROPOMI OFFL TOC with global ground-based measurements during the period from April to November 2018, showed a mean bias of 0 % to 1.5 % and standard deviations between 2.5 % and 4.5 % for monthly mean collocations (Garane et al., 2019).

A scientific S5P/TROPOMI product generated with the WFDOAS v4 algorithm was also used. The WFDOAS set up is identical to WFFA described above except for the narrower wavelength window (325-335 nm) and a third-degree polynomial used (Eq. (1)). Furthermore, WFDOAS uses temperature profiles reported with the ozone profile climatology rather than reanalysis data as in WFFA. Figure 3 shows a comparison of S5P/TROPOMI WFDOAS results with daily ground-based measurements between November 2017 and September 2019. S5P/TROPOMI-WFDOAS shows a bias of 2.0 % with $1\sigma$ of 1.9 % for Brewer instruments, and 2.1 % bias with 2.3 % standard deviation for Dobson instruments.

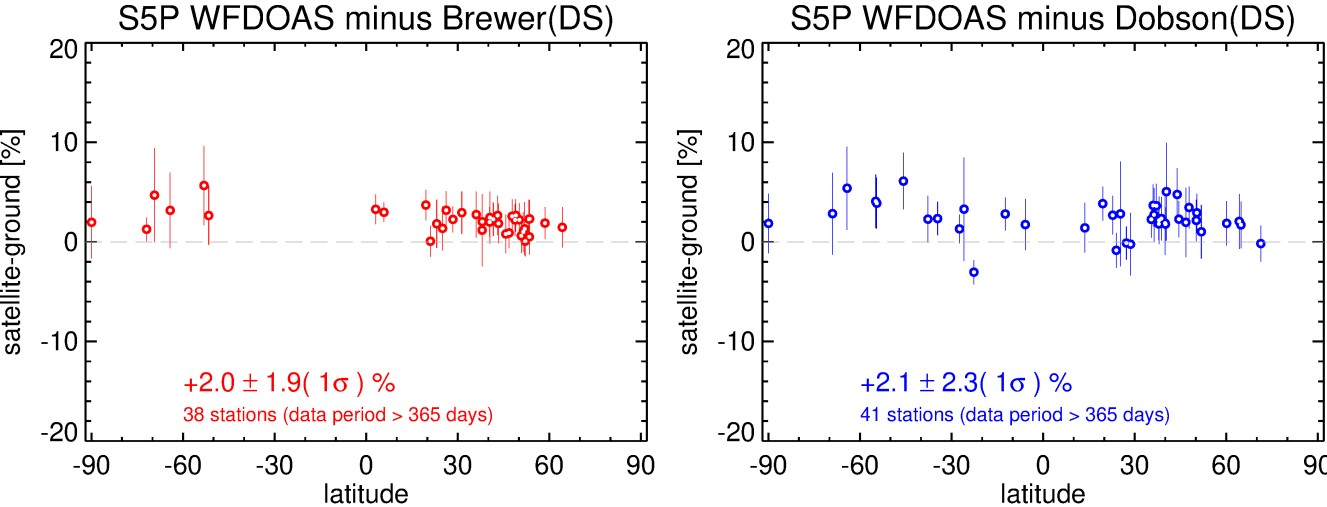

**Figure 3.** Summary of the daily mean comparison between ground-based measurements and S5P/TROPOMI WFDOAS TOC for Brewer (left) and Dobson (right) instruments.

To perform the comparison with ground-based data and between the S5P products, both datasets, OFFL and WFDOAS, have been binned into $0.3°\times0.3°$ boxes and averaged daily. These gridded data were also used for the comparison with OMPS-WFFA retrieval. Figure 4 shows the latitude-time comparison between TROPOMI WFDOAS and OFFL, exhibiting a global mean

difference of 1.5 % with 0.7 % standard deviation, with WFDOAS being higher than OFFL. Almost no seasonal variability is observed in the differences, larger differences occur in the southern hemisphere polar region during winter/spring.

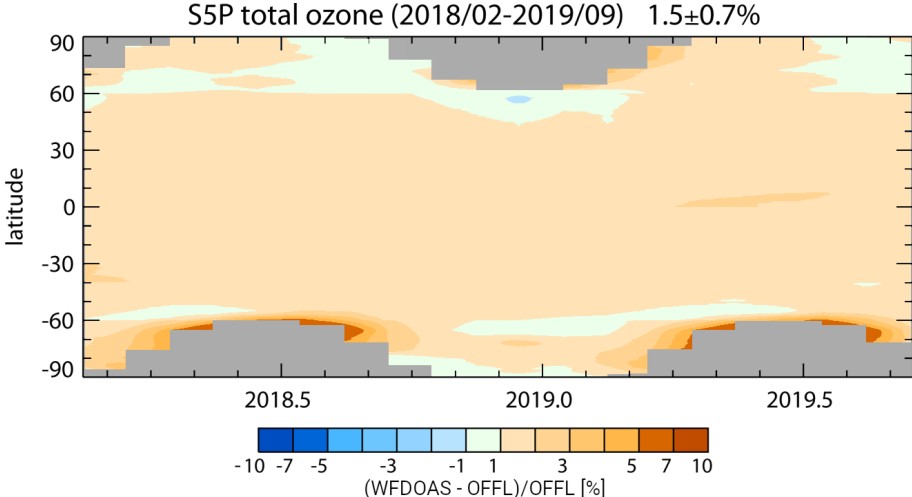

**Figure 4.** Latitude-time comparison between S5P/TROPOMI OFFL and S5P/TROPOMI WFDOAS total ozone from February 2018 to September 2019.

The S5P-WFDOAS product is retrieved using the Serdyuchenko et al. (2014) ozone absorption cross-sections. For the WFDOAS approach, the use of the Bass-Paur (BP, shifted by 0.23nm) and BDM ozone absorption cross-sections (Paur and Bass, 1985; Malicet et al., 1995) leads to retrieved total ozone being lower by $2-3$ %. We note that, the WFFA approach with

225 a wider spectral window and subtraction of a low order polynomial is weakly sensitive to the use of different ozone absorption cross-sections.

## 6   Validation of WFFA total ozone column

Three years (2016-2018) of OMPS/WFFA TOC data were daily averaged and gridded onto a $0.5° \times 0.5°$ grid, to perform the analysis and compare with other products. For the validation, percentage differences with respect to comparison datasets were

230 calculated as follows: $(WFFA - comparison\_data)/comparison\_data \cdot 100$.

Figure 5 shows seasonal maps of WFFA TOC for the analyzed period. The total ozone generally shows a minimum in the tropical region in all seasons. The meridional gradient of TOC is stronger during winter and spring for both hemispheres. In the subpolar region of the northern hemisphere, increased ozone values are observed during DJF and MAM. In the southern hemisphere, over the subpolar region, the maximum in TOC during austral spring (SON) is weaker than its counterpart in

the northern hemisphere. The minimum over the Antarctic during austral spring ("ozone hole") is observed. Over complex topography areas, like the Himalayas in Asia and the Andes in South America, lower ozone amounts are observed.

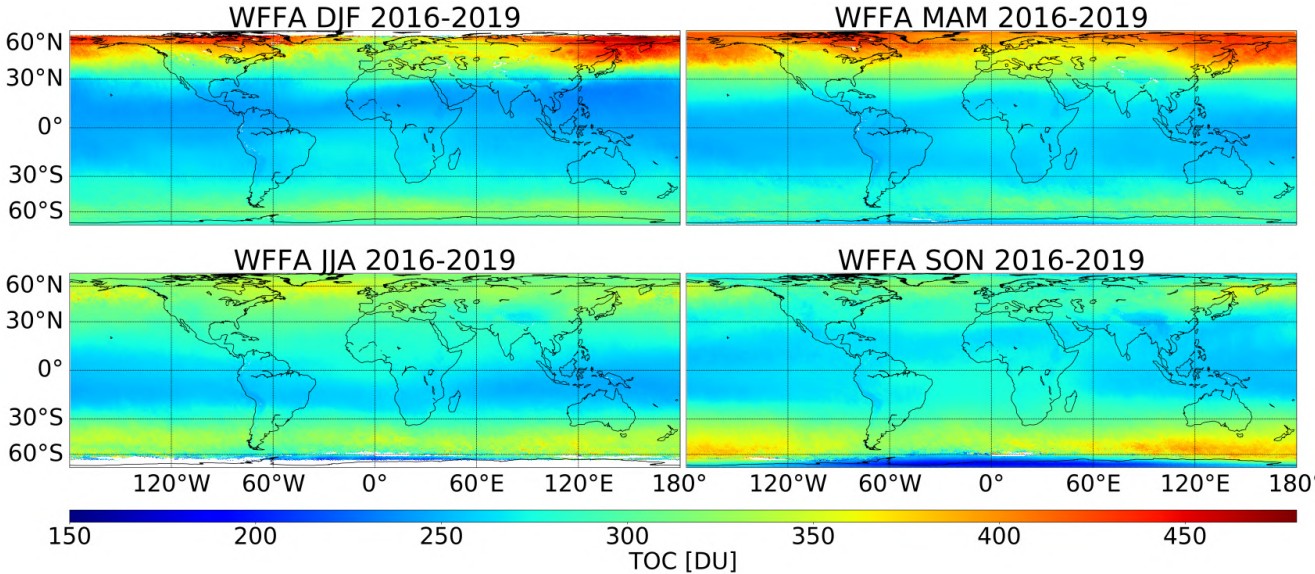

**Figure 5.** Seasonal average maps of WFFA/OMPS-NM total ozone column.

## 6.1 Comparison with ground-based measurements

Daily mean ground-based data for 48 stations were compared with daily satellite data averaged in the grid box that contains the station. Since only cloud-free satellite ground pixels were retrieved, the number of co-located days to be compared at a given station is rather low. Only stations with co-located data of at least 70 days were selected to have a sufficient sample for the comparison. With these criteria, 18 Dobson and 30 Brewer stations were available for the validation during the analyzed period.

Daily relative differences between WFFA TOC and the ground-based data were calculated. The mean relative differences vary from -2 % for Rio Gallegos (Brewer; 51.60° S, 69.32° W) to 4.8 % for Mauna Loa (Brewer; 19.53° N, 155.57° W). The high bias with respect to Mauna Loa data might result from the station's high altitude (3.4 km), while the grid box's average surface height is much lower (1.0 km). The standard deviation varies from 0.8 % for Paramaribo (Brewer; 5.81° N, 55.21° W) to 6.6 % for Amberd (Dobson; 40.38° N, 44.25° E). Figure 6 shows the time series and the relative differences for two selected stations as an example of the comparison, Santa Cruz (Brewer; 28.42° N, 16.26° W) and Tamanraseet (Dobson; 22.80° N, 5.52° E). Figure 6 shows that the seasonality of both WFFA and ground-based data is similar. A very good agreement in the seasonality and the TOC values are observed for all considered ground stations. From a total of 48 stations, 28 show a bias of less than 1 % and 27 stations show a standard deviations of less than 3 %.

Figure 7 presents the summary of the comparisons with Brewer (left) and Dobson instruments (right) as a function of latitude. A distinction between the instruments was made because they might show differences of up to 4 % in their direct sun measurements (Feister, 1994; Vanicek, 2006). Overall, the bias between WFFA and ground-based measurements is positive, 0.2

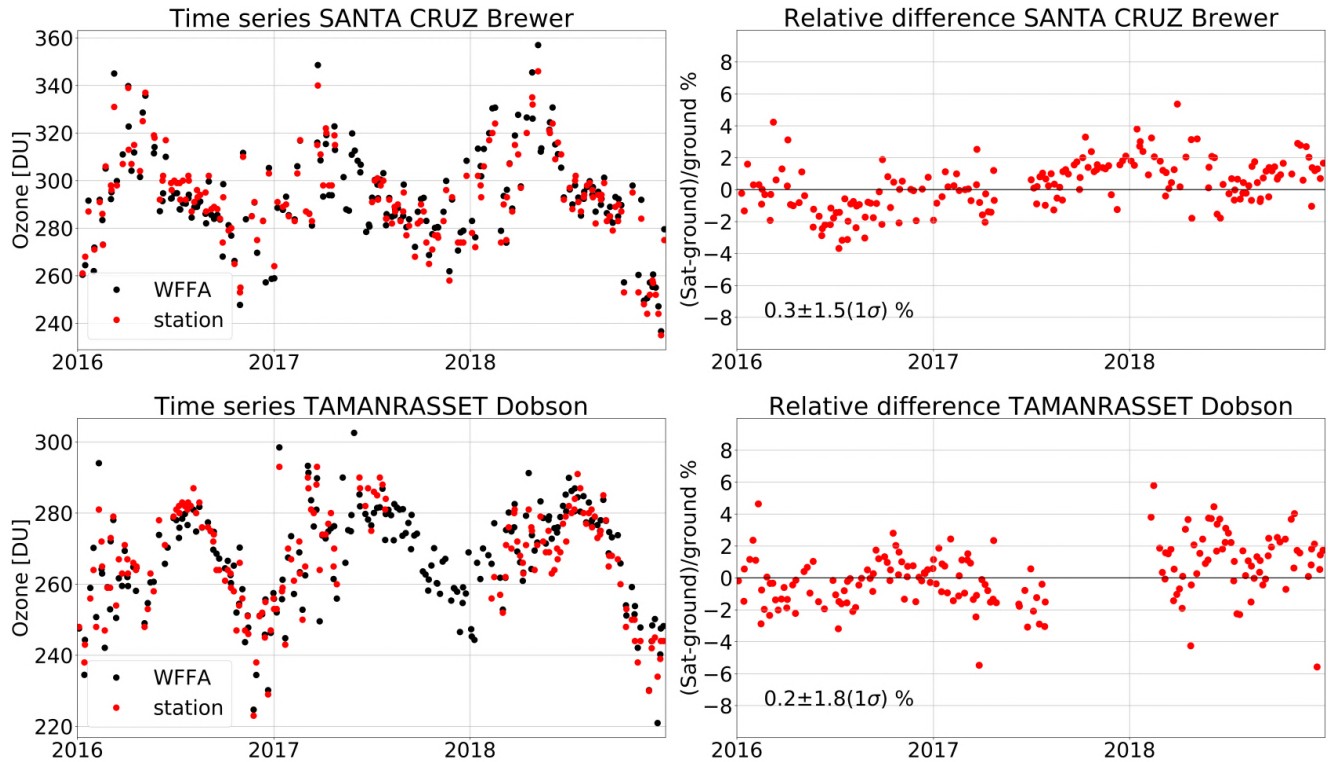

**Figure 6.** Left panels: Examples of daily mean total ozone time series from ground-based measurements (red) and co-located WFFA TOC (black) from 2016 to 2018, in Santa Cruz (28.42° N, 16.26° W) and Tamanraseet (22.80° N, 5.52° E). Right panels: Percentage differences between WFFA and ground-based data. Mean relative difference and its standard deviation are indicated.

% for Brewer and 0.5 % for Dobson instruments, with a mean standard deviation of 1.3 %. For stations with both instruments, Athens (37.98° N, 23.73° E) and Tamanraseet (22.80° N, 5.52° E), the differences between Dobson and Brewer are 1.7 % and 0.5 %, respectively. No particular patterns between hemispheres are observed. Averaging all stations, WFFA TOC exhibits a mean bias of $0.3\pm1.3(1\sigma)$ %.

## 6.2 Comparison with OMPS-NM operational product and S5P/TROPOMI

WFFA results have been compared to the operational total ozone column product of OMPS-NM L2 v2.1 (OMPS-L2), and two different retrievals from S5P/TROPOMI (OFFL and WFDOAS) as introduced in Section 5.

A comparison for one orbit on June 10, 2018, is shown in Fig. 8. The upper panels show the TOC of the central across-track FOV (18) against latitude and SZA for all datasets. The lower panels show the percentage differences of WFFA results with respect to the comparison datasets. The ozone total column reaches a minimum in the tropics increasing towards the poles, with local maxima at 40° S and 70° N. The absolute maximum is observed at 50° N. All satellite data show very good agreement in the variation of TOC with latitude and SZA. The mean bias with respect to OMPS-L2 is 0.39 %. The differences with

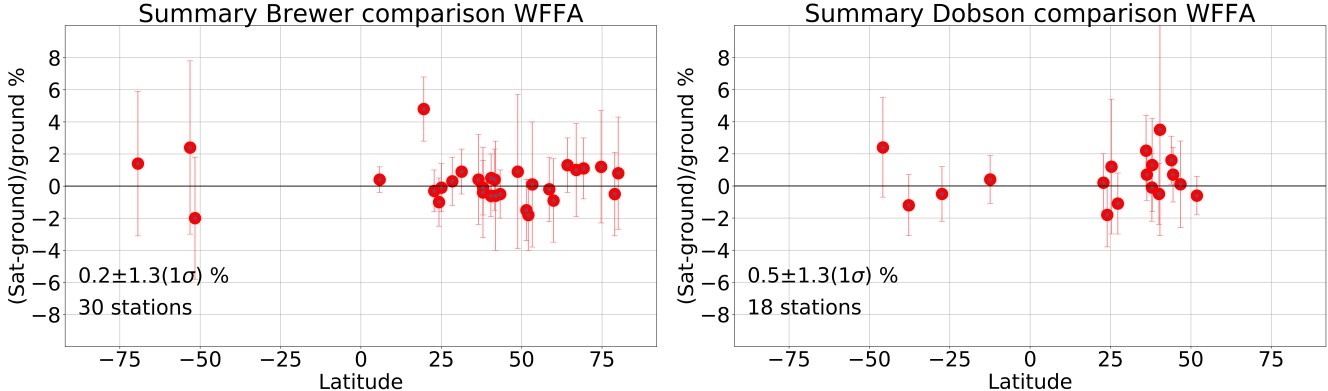

**Figure 7.** Summary of the mean relative differences between WFFA results and ground-based measurements for Brewer (left) and Dobson (right) instruments from 2016 to 2018. Mean differences and their standard deviations are indicated along with the number of stations analyzed.

respect to S5P OFFL and WFDOAS data, are -0.36 % and -2.48 %, respectively. S5P WFDOAS exhibits more ozone than the other datasets along the entire orbit. This is expected considering the direct comparisons between the two S5P datasets shown above (Section 5.3). Between -70° to 40° SZAs (approximately 40° S to 60° N in latitude), differences with respect to OMPS L2 and S5P OFFL data vary around ±1 %. For larger SZAs, WFFA results differ by less than 2 % with respect to the three comparison datasets, except for the first pixel of the considered orbit. A difference between hemispheres is observed, for the northern hemisphere WFFA shows more ozone than S5P OFFL and OMPS-L2, while for the southern hemisphere WFFA TOC is lower. The standard deviations of the differences are similar for all three comparison datasets, varying between 1.1 % for OMPS-L2 and 1.4 % for S5P WFDOAS.

To carry out a more general comparison, by looking at seasonal and global averages, the three comparison datasets were gridded in the same way as WFFA data. For OMPS-L2 the same orbits and ground pixels as those for WFFA were selected (ground pixels with cloud fraction less than 0.1, SZA smaller than 80° and only across-track FOVs from 10 to 22), from 2016 to 2018. For S5P all available data (all FOVS as well as cloudy scenes included) were gridded. The regular production of the OFFL data started on April 30, 2018. To compare an entire 12 month period, WFFA TOC was retrieved until May 2019. Thus, the comparison with S5P/TROPOMI OFFL and WFDOAS comprised data from June 2018 until May 2019. The comparison was only made for daily grid boxes with data available for WFFA.

Figure 9 shows maps of seasonal relative deviations of WFFA results to those from OMPS-L2 (left) and S5P OFFL (right). In general, WFFA has a positive bias with respect to OMPS L2 and a negative with respect to S5P OFFL. Larger differences are observed in the polar regions. During austral autumn and winter (MMA and JJA) WFFA TOC is lower than the other two satellite datasets in the polar region, while during the austral summer (DJF) is higher. Over areas with complex topography, like the Himalayas in Asia, the Great Rift Valley in Africa, and the Andes in South America, WFFA ozone values are larger than OMPS-L2 by up to 6 % but are in good agreement with S5P OFFL. As it was seen in Fig. 5, WFFA shows lower ozone

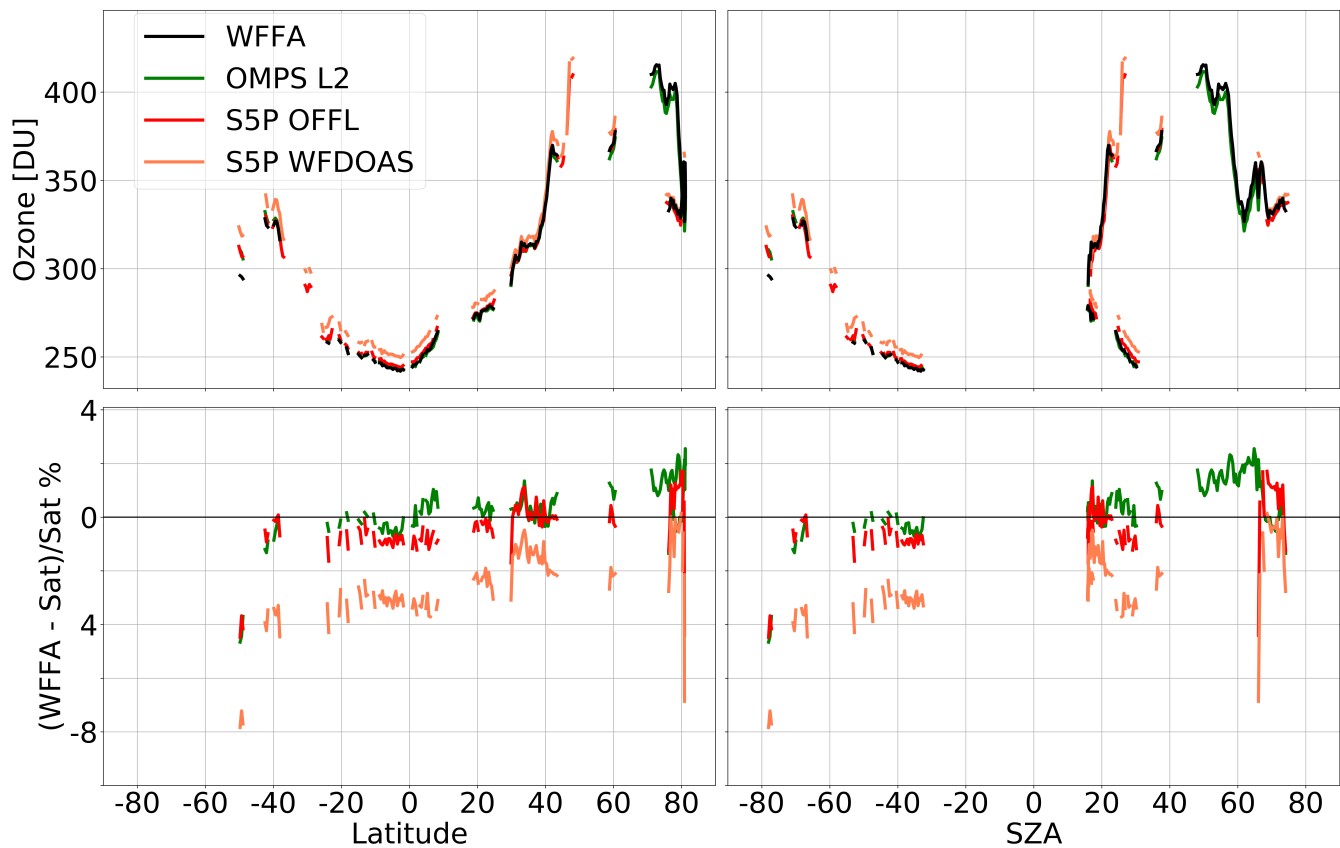

**Figure 8.** Ozone total column (top) and percentage differences (bottom) for an example orbit, against latitude (left) and SZA (right), for the central FOV (18) of the orbit. OMPS orbit number 34298, on the 10[th] of June 2018. Southern hemispheric SZA values are plotted as negative numbers for clarity.

for scenes with high surface elevation than in the surrounding areas, the same was observed for OMPS-L2 (not shown) with even lower values than WFFA, which explain the larger differences over, for example, the Andes.

From the differences of WFFA with respect to OMPS-L2, a positive bias over both poles, and a bias of around 4 % in southern subtropics and at northern mid-latitudes are observed during boreal winter. Globally a mean positive bias of $0.6\pm1.5(1\sigma)$ % is observed. During boreal spring, the bias dissipates in the southern subtropics and becomes less persistent at northern mid-latitudes. Combined with larger negative differences in the southern polar area, this results in a global mean bias of $0.2\pm1.3(1\sigma)$ % for MAM. In boreal summer, a 2 % bias is observed in the northern subtropics, decreasing in autumn (SON). The higher bias in the summer hemisphere's subtropical areas is possibly related to the Inter-Tropical Convergence Zone (ITCZ). Although only cloud-free scenes are retrieved, some of the ground pixels may still be contaminated by clouds, which might result in small systematic biases. The yearly global mean difference is $0.0\pm1.3$ %.

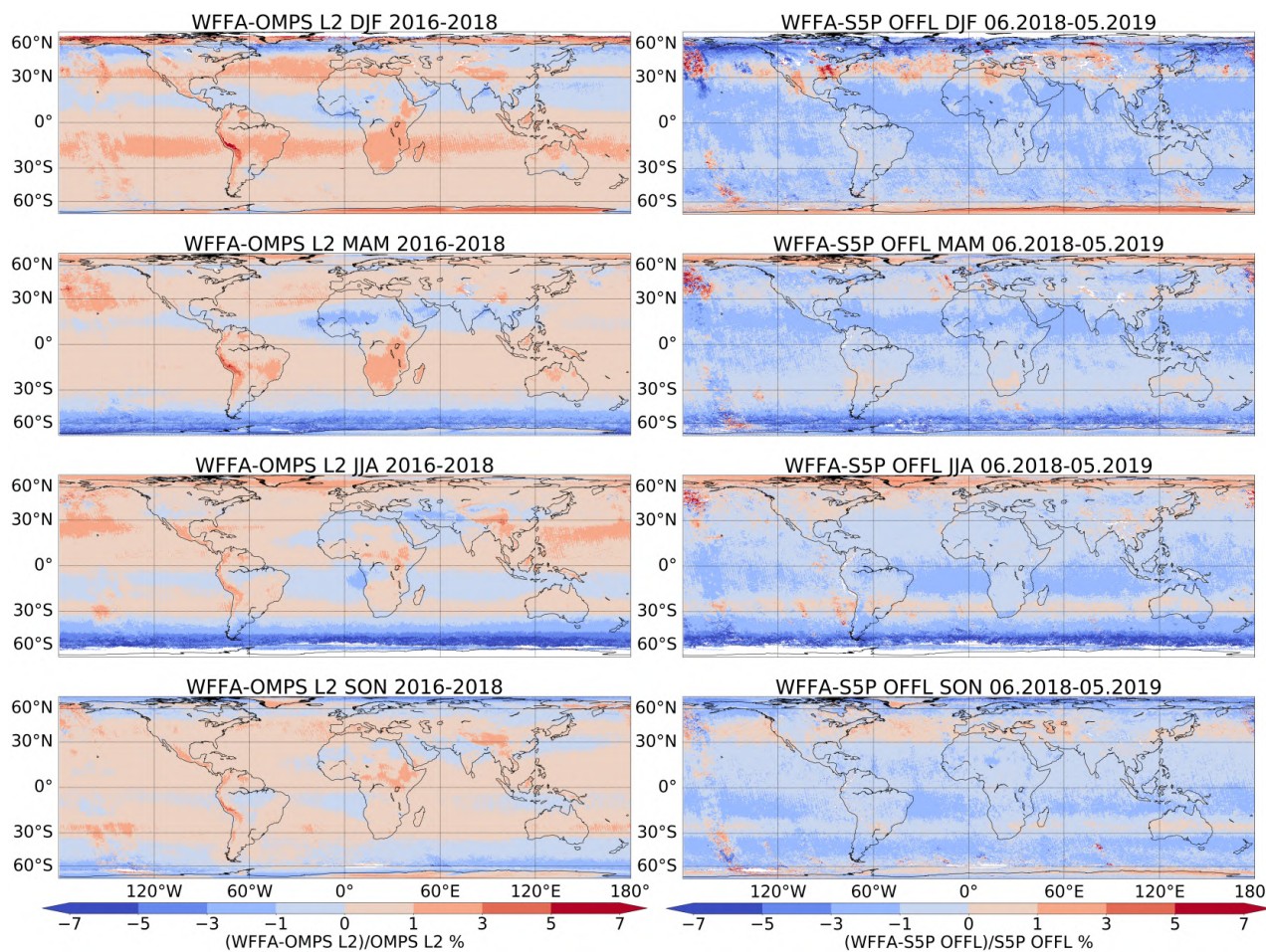

**Figure 9.** Relative differences in the total ozone column between the seasonally averaged WFFA data and two other satellite's products. Left panel: relative differences with respect to OMPS-L2. Right panel: relative differences with respect to S5P OFFL.

The comparison between WFFA and S5P/TROPOMI results is shown in the right panels of Fig. 9. Striping is seen in the differences to S5P most likely due to differences in the grid boxes' sampling. For S5P, the topography distinction is seen over
the Andes and the Himalayas, only during boreal winter and spring. Similar patterns to those observed for OMPS L2 are seen over the polar regions, except in the northern pole during boreal winter, where S5P OFFL TOC is up to 4 % higher than WFFA. The subtropical positive bias band observed for OMPS-L2 is negative and within 1 % for S5P OFFL. For areas where WFFA TOC is less than OMPS-L2 TOC, like over southern subtropics during austral winter, S5P OFFL shows even higher values. The global mean relative differences with respect to S5P OFFL are -0.6±1.5(1$\sigma$) for DJF, -0.8±1.5(1$\sigma$) for MAM, -0.8±1.2(1$\sigma$)
for JJA, and -0.8±1.5(1$\sigma$) for SON.

For a more detailed analysis, TOC time series for five zonal bands were calculated: high northern latitudes (60°-90° N), northern mid-latitudes (30°-60° N), tropics (30° N-30° S), southern mid-latitudes (30°-60° S), and southern high latitudes (60°-

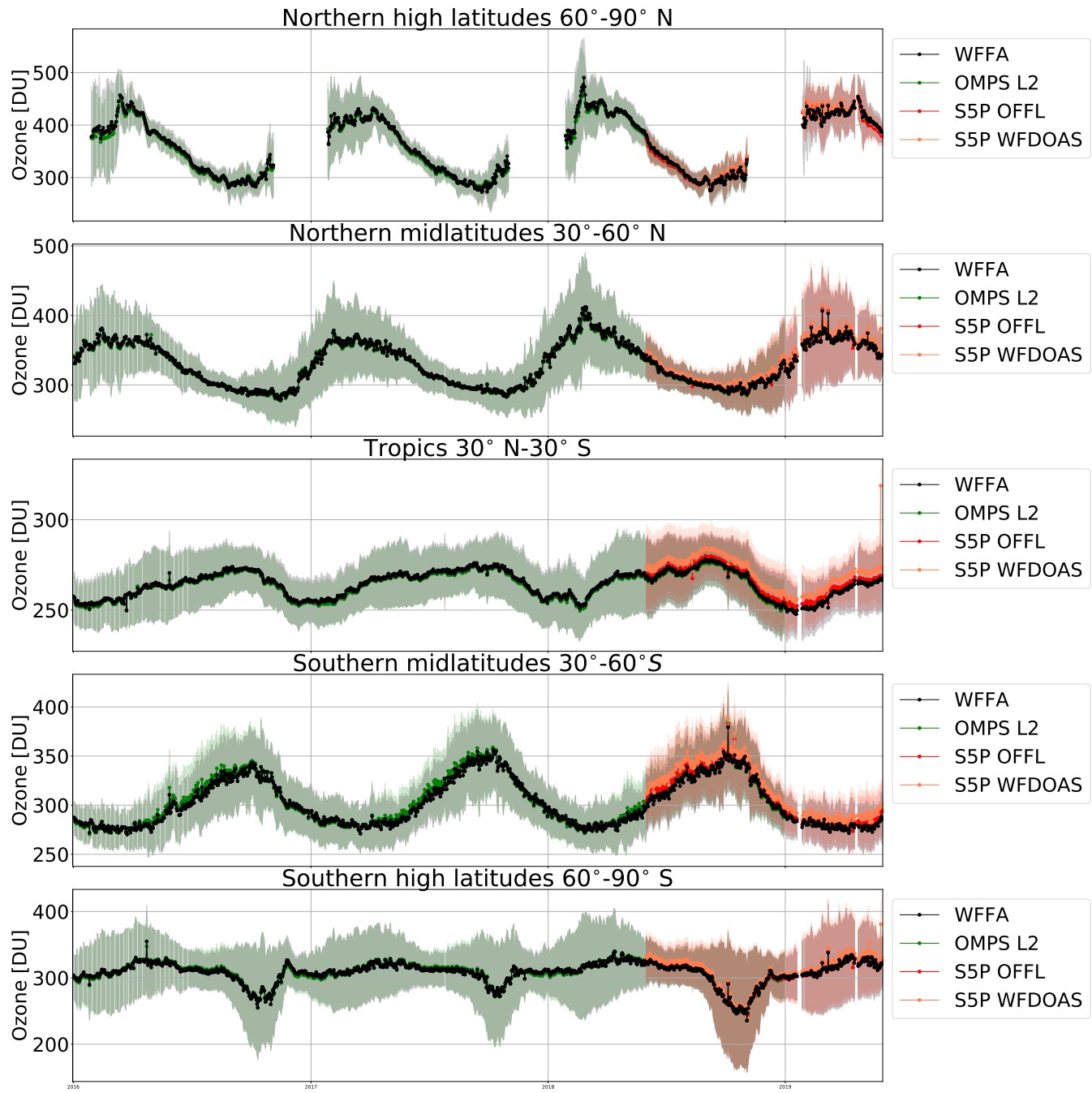

**Figure 10.** Zonal mean time series of WFFA, OMPS-L2, S5P OFFL and S5P WFDOAS TOC, for five latitudinal bands. The shading indicate the standard deviations of the time series.

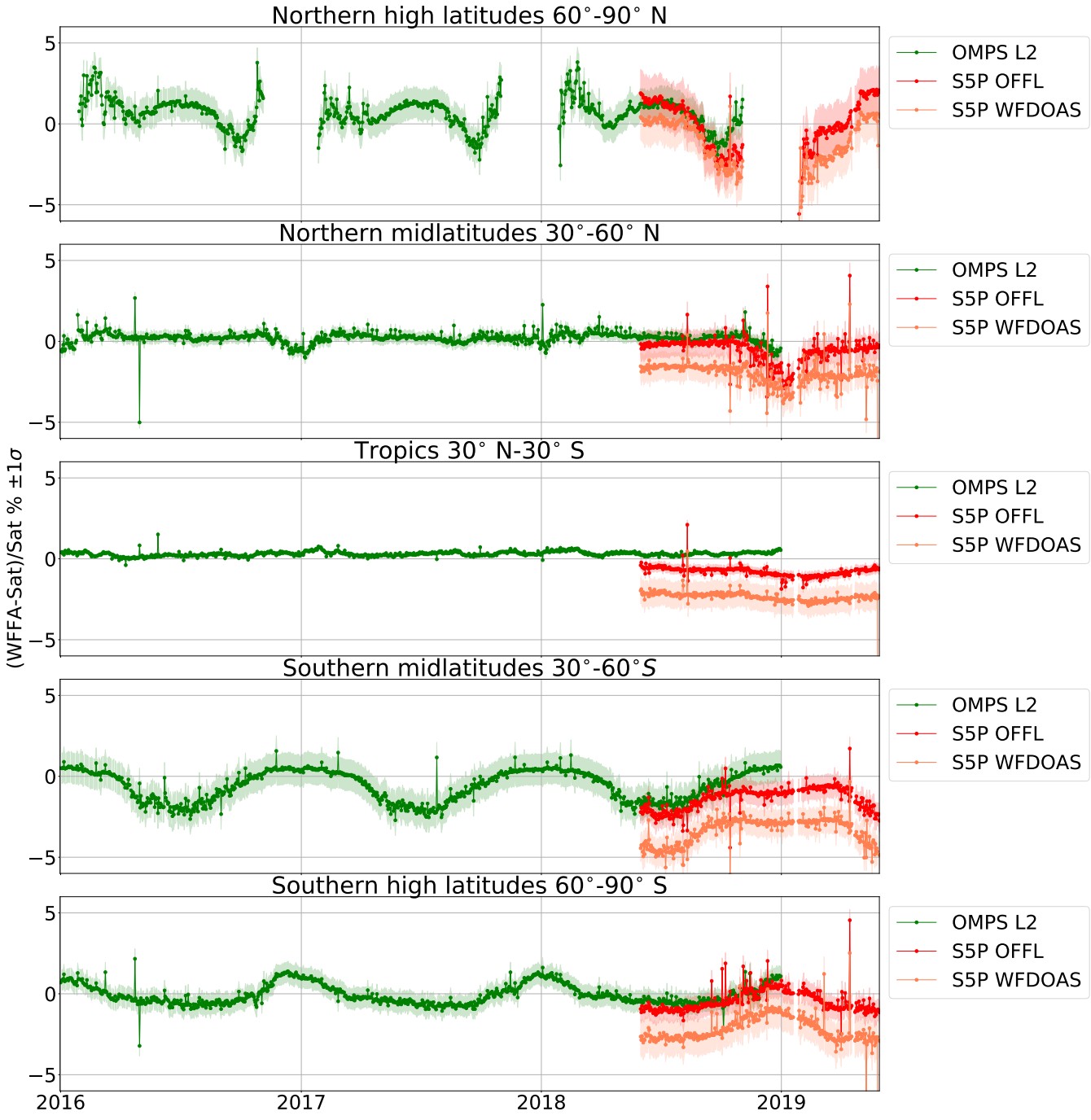

**Figure 11.** Differences of the zonal mean time series of WFFA with respect to the results from OMPS-L2, S5P OFFL and S5P WFDOAS, for five latitudinal bands. The shading indicate the standard deviations of the time series.

90° S), as shown in Fig. 10. The mean relative differences in these zonal bands are shown in Fig. 11 and summarised in Table 2. In general, the four different datasets follow the same seasonality and short-term variability, generally showing very good

agreement. However S5P products, OFFL and WFDOAS, are typically higher than OMPS-L2 and WFFA, particularly higher in the tropics and in the southern mid-latitudes. A persistent mean negative bias is observed with respect to S5P WFDOAS as it was seen in the comparison for one sample orbit in Fig. 8.

Figure 11 shows larger variations at high northern latitudes, particularly during boreal winter. Nevertheless, the mean differences in the 60°-90° N band, are 0 % with respect to S5P-OFFL, and less than 1.2 % for the other datasets. At northern

mid-latitudes, WFFA shows a bias of 0.2 % with respect to OMPS-L2, -0.5 % with respect to S5P-OFFL, and -2.0 % with respect to S5P-WFDOAS. In the tropics, the differences between the datasets are fairly constant with time, with biases of 0.3 % for OMPS-L2, -0.8 % for S5P-OFFL and -2.4 % for S5P-WFDOAS; the standard deviations are below 0.8 %. At southern mid-latitudes, WFFA shows less ozone than OMPS-L2 during winter, by about -3 %. The relative difference decreases in autumn and spring and becomes slightly positive during the summer. The same pattern is observed when comparing with S5P, with the

mean relative differences ranging from -1.4 for OFFL to -3.4 % for WFDOAS. At high southern latitudes, WFFA results show similar seasonal behaviour as in the mid-latitudes. Overall there is a -0.1 % bias with respect to OMPS-L2, and the standard deviation is 0.6 %($1\sigma$). Very good agreement (bias -0.5 %) of both WFFA and OMPS-L2 with S5P-OFFL is observed at these latitudes.

**Table 2.** Relative differences and standard deviations between WFFA/OMPS-NM and OMPS L2, S5P/TROPOMI OFFL and S5P/TROPOMI WFDOAS in various zonal bands.

| Dataset | 90°-60° N | 60°-30° N | 30° N-30° S | 30°-60° S | 60°-90° S |
|---|---|---|---|---|---|
| OMPS L2 (2016-2018) | $0.6 \pm 0.9\%$ | $0.2 \pm 0.4\%$ | $0.3 \pm 0.1\%$ | $-0.6 \pm 0.9\%$ | $-0.1 \pm 0.6\%$ |
| S5P OFFL (06.2018-05.2019) | $0.0 \pm 1.5\%$ | $-0.5 \pm 0.8\%$ | $-0.8 \pm 0.3\%$ | $-1.4 \pm 0.7\%$ | $-0.5 \pm 0.7\%$ |
| S5P WFDOAS (06.2018-05.2019) | $-1.2 \pm 1.4\%$ | $-2.0 \pm 0.8\%$ | $-2.4 \pm 0.8\%$ | $-3.4 \pm 0.8\%$ | $-2.2 \pm 1.1\%$ |

## 7    Summary and conclusions

In this study we present a new scientific TOC product from OMPS-NM observations using the WFFA technique, which is a modified retrieval approach adapted from the WFDOAS algorithm. A new ozone profiles climatology was generated for the retrieval, using OMPS-LP profiles (Arosio et al., 2018) and ozonesondes.

OMPS-WFFA data was validated using ground-based measurements from the WOUDC dataset and three other TOC satellite datasets: OMPS-NM Level 2, S5P/TROPOMI OFFL and S5P/TROPOMI WFDOAS. The comparison with ground-based

measurements shows a mean bias below 1 % for 28 of a total of 48 stations. For 27 stations, the standard deviations of the mean differences are under 3 %. In total, a mean bias of +0.3 % and a standard deviation of 1.3 % were found. These values

are similar to those reported by the operational product of OMPS-NM and by S5P/TROPOMI (Section 5). All comparisons between WFFA TOC and other satellite products are consistent, concerning seasonality and variability with latitude. WFFA TOC presents a zero yearly global mean bias with respect to OMPS-L2, approximately -0.6 % with respect to S5P OFFL and -2.5 % with respect to S5P WFDOAS. The standard deviations of the differences are around 1.4 % for all satellite validation datasets. Larger differences were found for polar regions and larger SZAs.

The newly created WFFA OMPS-NM total ozone dataset is intended to be used for retrieving tropospheric ozone columns employing the limb-nadir matching technique in combination with OMPS-LP data.

*Data availability.* The ozone and temperature climatology is available at:

http://www.iup.uni-bremen.de/UVSAT/datasets/iup-ozone-profile-climatology

## Appendix A: Retrieval development and sentitivity tests

### A1 From WFDOAS to WFFA

The use of the original WFDOAS approach in the typical spectral window (325 nm to 335 nm) with OMPS-NM data, results in large variations of the retrieved TOC for different across-track ground pixels. Fig. A1 shows ozone anomalies (TOC value minus the mean of all across-track FOVs) for all the orbits of one day, averaged over the tropics (10° S - 10° N) as a function of the across-track index. The left panel shows the results from the original WFDOAS approach, using a cubic polynomial and the spectral window 325 to 335 nm. There are systematic differences between ground pixels, for instance, more than 5 DU differences between FOV 18 and 19. The results using a cubic polynomial and wider window (316 to 336 nm) are shown in the central panel. For this configuration, the differences between adjacent pixels are smaller, but a large variation of about 30 DU is observed from the first to the last across-track FOV. The right panel shows the results from the WFFA approach, using a constant instead of the cubic polynomial and the fitting window from 316 to 336 nm. The variation between pixels is below 2 DU for across-track FOVs 10 to 22.

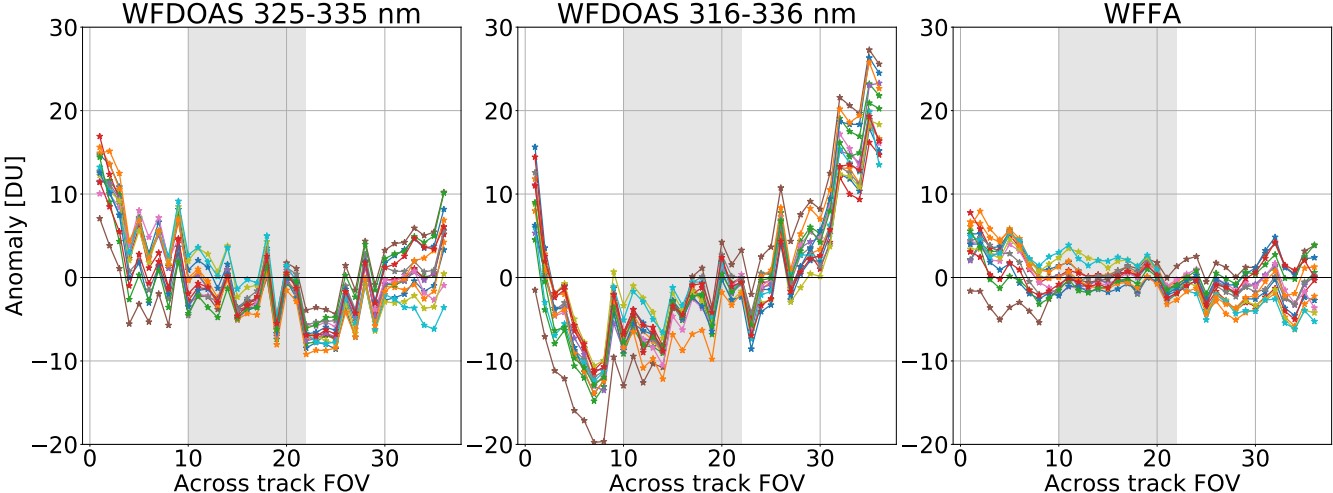

**Figure A1.** Tropical averaged ozone anomalies for all orbits of one day (10[th] of January 2018) for different configurations of the retrieval. The grey shading indicates the FOVs used in this study. Left panel: original WFDOAS. Central panel: WFDOAS with a wider spectral window. Right panel: WFFA.

### A2 All, even and odd spectral points

Table A1 shows the wavelengths used in the fit for the central across-track FOV (18) for the cases of all, even and odd spectral points. Since the non-linear fit adjust the earthshine spectrum's wavelength grid to the solar reference spectrum, the final wavelength grid is different for every across-track FOV.

**Table A1.** Wavelengths processed in the retrieval for the FOV 18 in the cases of all, even or odd-numbered spectral points in nm.

| All | Even | Odd | All | Even | Odd | All | Even | Odd |
|---|---|---|---|---|---|---|---|---|
| 316.1672 | | 316.1672 | 322.8542 | | 322.8542 | 329.5337 | | 329.5337 |
| 316.5854 | 316.5854 | | 323.2719 | 323.2719 | | 329.9509 | 329.9509 | |
| 317.0036 | | 317.0036 | 323.6895 | | 323.6895 | 330.3682 | | 330.3682 |
| 317.4217 | 317.4217 | | 324.1071 | 324.1071 | | 330.7854 | 330.7854 | |
| 317.8398 | | 317.8398 | 324.5247 | | 324.5247 | 331.2026 | | 331.2026 |
| 318.2579 | 318.2579 | | 324.9423 | 324.9423 | | 331.6198 | 331.6198 | |
| 318.6759 | | 318.6759 | 325.3598 | | 325.3598 | 332.0370 | | 332.0370 |
| 319.0939 | 319.0939 | | 325.7773 | 325.7773 | | 332.4541 | 332.4541 | |
| 319.5118 | | 319.5118 | 326.1948 | | 326.1948 | 332.8712 | | 332.8712 |
| 319.9298 | 319.9298 | | 326.6122 | 326.6122 | | 333.2884 | 333.2884 | |
| 320.3476 | | 320.3476 | 327.0296 | | 327.0296 | 333.7055 | | 333.7055 |
| 320.7655 | 320.7655 | | 327.4470 | 327.4470 | | 334.1226 | 334.1226 | |
| 321.1833 | | 321.1833 | 327.8644 | | 327.8644 | 334.5396 | | 334.5396 |
| 321.6011 | 321.6011 | | 328.2817 | 328.2817 | | 334.9567 | 334.9567 | |
| 322.0188 | | 322.0188 | 328.6991 | | 328.6991 | 335.3737 | | 335.3737 |
| 322.4366 | 322.4366 | | 329.1164 | 329.1164 | | 335.7908 | 335.7908 | |

The selection of the odd-numbered spectral points was made after investigating the effect of the various wavelength choices on the retrieval. When using odd-numbered wavelengths, the retrieval result does not change much whether the temperature fit parameter is included in the fit or not. For the same orbits as used in the previous section (Sec. A1), the retrieval was applied with and without fitting the temperature parameter for the three wavelength selections. Figure A2 shows the relative differences of the results with and without the temperature parameter. The left panel shows the average difference over the across-track FOVs, used in this study, as a function of latitude. The average over the tropics as a function of the across-track index is shown in the right panel. It is observed that the dependence on the temperature of the odd sample is significantly weaker both across and along-track. Below 40° N and for the central FOVs, the differences for the odd sample are less than 0.5%, while for the other two data sets, they are between 0.5% and 1%. In the standard WFFA retrieval using the odd-numbered wavelength sample, as shown in the main part of the paper, the temperature is still included in the fit procedure.

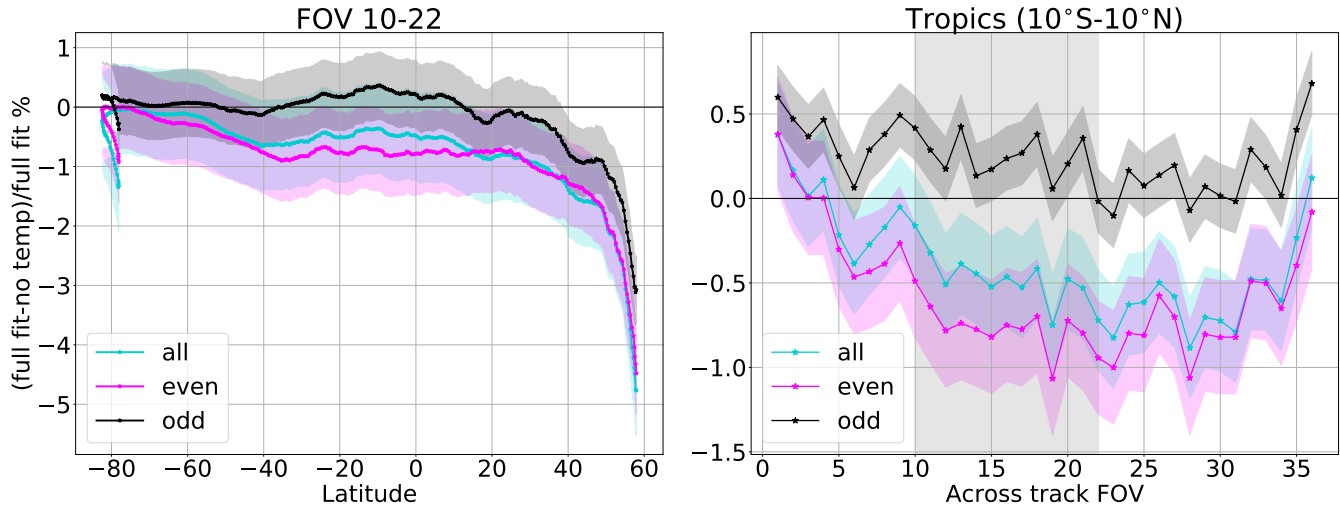

**Figure A2.** Mean relative differences between the results from the fits including and excluding the temperature, for the orbits of one day (10[th] of January 2018) for different wavelength samples used. Left panel: Average over the across-track FOVs 10 to 22 as a function of latitude. Right panel: Average over the tropics as a function of the across-track index. Standard deviations are shown by shadings.

## A3 Sensitivity to aerosols scenarios

**Table A2.** Results and errors for different boundary layer aerosol scenarios. The true value of TOC is 325 DU.

| Aerosol type | LER albedo | TOC | Error % | Aerosol type | LER albedo | TOC | Error % |
|---|---|---|---|---|---|---|---|
| Surface albedo = 0.05 - SZA 27.02° | | | | Surface albedo = 0.2 - SZA 27.02° | | | |
| Maritime | 0.164 | 334.3651 | 2.88 | Maritime | 0.263 | 334.3529 | 2.89 |
| Rural | 0.196 | 335.0037 | 3.08 | Rural | 0.263 | 334.6262 | 2.96 |
| Tropospheric | 0.216 | 335.5622 | 3.25 | Tropospheric | 0.286 | 335.0233 | 3.08 |
| Urban | 0.081 | 329.3202 | 1.33 | Urban | 0.110 | 328.8105 | 1.17 |
| Surface albedo = 0.05 - SZA 59.88° | | | | Surface albedo = 0.2 - SZA 59.88° | | | |
| Maritime | 0.286 | 326.5662 | 0.48 | Maritime | 0.349 | 326.3276 | 0.41 |
| Rural | 0.295 | 325.8940 | 0.28 | Rural | 0.323 | 325.5785 | 0.18 |
| Tropospheric | 0.335 | 326.0780 | 0.33 | Tropospheric | 0.367 | 325.7167 | 0.22 |
| Urban | 0.062 | 322.3880 | -0.80 | Urban | 0.066 | 322.3423 | -0.82 |

**Table A3.** Results and errors for extreme volcanic stratospheric aerosol loading . The true value of TOC is 325 DU.

| LER albedo | TOC | Error % | LER albedo | TOC | Error % |
|---|---|---|---|---|---|
| Surface albedo = 0.05 - SZA 27.02° | | | Surface albedo = 0.2 - SZA 27.02° | | |
| 0.205 | 348.9609 | 7.37 | 0.294 | 349.5970 | 7.57 |
| Surface albedo = 0.05 - SZA 59.88° | | | Surface albedo = 0.2 - SZA 59.88° | | |
| 0.325 | 328.1177 | 0.96 | 0.381 | 328.5581 | 1.09 |

The Lambertian Equivalent Reflectivity (LER) effective scene albedo represents a first-order correction for non-absorbing aerosols. For the WFDOAS technique, the ozone might be underestimated by 1 % in the presence of absorbing aerosols with a visibility of 2 km (Coldewey-Egbers et al., 2003). Since the WFFA approach is slightly different from WFDOAS, similar sensitivity tests using different aerosol scenarios were performed to confirm the prior results.

We generated synthetic radiances for different aerosol scenarios, using SCIATRAN V4.2 with the aerosol parametrization from LOWTRAN (Kneizys et al., 1986; Shettle and Fenn, 1979). From these radiances, the LER albedo was retrieved and used in the WFFA retrieval. The synthetic radiances were calculated with a total ozone of 325 DU, solar zenith angles of 59.88° and 27.02° (chosen from real values of OMPS-NM ground pixels), visibility of 2 km and surface albedos of 0.05 and 0.2. The different types of boundary layer aerosols are maritime, rural, tropospheric and urban. One case with extreme volcanic stratospheric aerosol loading was included. The results are summarized in Table A2 for the boundary layer scenarios and in Table A3 for the stratospheric loading.

For large SZAs, the aerosol effect is largely accounted for with the effective scene albedo, particularly for weakly absorbing boundary layer aerosols (urban). In the case of strongly absorbing boundary layer aerosols, uncertainties are somewhat larger but still within 1%. For small SZAs, the retrieved TOC might be overestimated by about 3 % for weakly absorbing aerosols and by 1 % for strongly absorbing aerosols. In case of an extreme volcanic aerosol loading in the stratosphere, the retrieved TOC might be overestimated by about 8 % for small SZAs, and by about 1 % for large SZAs.

## A4   Sensitivity to tropospheric ozone

To investigate the sensitivity of the retrieval to the tropospheric ozone amount, we scaled the lower part of the climatological ozone profiles (below 12 km) by factors 2 and 5 and repeated the retrieval. At each iterative step the ozone profile to be used in the forward model is extracted from the climatology in accordance with the total ozone column value obtained at the previous iteration (300 DU for the first iteration) and its lower part is scaled as described above. No significant differences in the retrieved total ozone value were identified.

*Author contributions.* All authors contributed to the design of the study. Andrea Orfanoz-Cheuquelaf developed the retrieval algorithm, performed the computer calculations and made the comparisons supervised by Mark Weber, Alexei Rozanov and Annette Ladstätter-Weißenmayer with John P. Burrows providing scientific conceptual input and oversight. Carlo Arosio provided vertical ozone profiles used in the study from inversions of OMPS-LP observations. Andrea Orfanoz-Cheuquelaf led the preparation of the manuscript. All authors contributed to the writing, editing and and evolution of the manuscript.

*Competing interests.* The authors declare that they have no conflict of interest.

*Acknowledgements.* This research has been funded in parts by the University and the State of Bremen, in parts by ESA-Ozone-CCI+, BMBF SynopSys-Ozone, and the German Research Fundation (DFG) through the research unit VolImpact (FOR2820) and project VolARC. Carlo Arosio acknowledges the support by the PRIME programme of the German Academic Exchange Service (DAAD) via funds from the German Federal Ministry of Education and Research (BMBF) and ESA's Living Planet Fellowship SOLVE. Most of the calculations reported here were performed on HPC facilities of the IUP, University of Bremen, funded under DFG/FUGG grant INST 144/379-1 and INST 144/493-1. The limb ozone profiles were processed on the German HLRN (High Performance Computer Center North). The GALAHAD Fortran Library was employed in the retrieval scheme.

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
