# Peer review of "Total ozone column from OMPS-NM measurements using the broadband Weighting Function Fitting Approach (WFFA)"

_Atmospheric Measurement Techniques, 2021_

## Author Comment (AC1)

**Review of "Total ozone column retrieval from OMPS-NM measurements" by Orfanoz-Cheuquelaf et al.**
**Anonymous Referee #2**

This paper presents a modified version of the WFDOAS algorithm to retrieve total ozone columns from the OMPS-NM instrument. The ultimate goal is to combine the nadir and the limb OMPS instruments to derive tropospheric ozone columns. In addition to the realized algorithmic developments, the study also investigates the quality of the retrievals by comparing them with ground-based data as well as with independent OMPS and TROPOMI retrievals.

The topic of this work fits well for AMTD and, from the presented results, the product quality appears to be good. However, I find that the section on the retrieval algorithm description currently lacks of details, justification and illustrations for the different choices that have been made. Details are provided below as major comments. I encourage consolidating the manuscript and to submit a revised version to AMTD.

We appreciate your comments.

After investigations of the spatial patterns of the limb-nadir matching for retrieving tropospheric ozone, we extended the total ozone retrieval to cover FOVs ranging from 10 to 22. Therefore, the following figures have changed: Figs. 5-7, 9 and 10. The added FOVs do not change the main conclusions in the manuscript.

All your minor comments have been addressed and will be included in the revised version. With respect to the major comments, we detail the following points:

**Major comments:**
*Retrieval algorithm:*

- The modifications made to the retrieval algorithm should be better justified and illustrated. I understand that the lower spectral resolution of OMPS may lead to more important cross-correlation and that there is a need to extent the fitting window. However, why considering one every two wavelengths only, which is expected to reduce the benefit of extending the fit window?

For ozone total column retrieval from OMPS-NM measurements, the spectral range extension to shorter wavelengths and reduction of the sampling work in the same direction, namely reducing the weight of the ozone differential absorption features in the retrieval and increasing the weight of the broad-band spectral signature of ozone absorption. This is the main objective of the algorithm changes which are justified by a differential structure of ozone absorption being strongly smoothed at the spectral resolution of OMPS-NM, that is significantly lower than that of GOME, GOME-2, OMI, or SCIAMACHY.

Could you illustrate the cross-correlation between fitted parameters for the different options (small and large windows, wavelength selection)?

With respect to the small vs large window, this is not a matter of correlation. The point is that using a larger window enables us to use a zeroth-order polynomial (i.e. constant) instead of the cubic one. This

increases significantly the information content and makes the retrieval much more robust. For the wavelength sampling, we investigated correlations between the weighting functions of ozone and temperature. For all cases, correlations are high and do not differ much between the cases. Having, however, in mind that also the spectral signature of the rotational Raman scattering is included in the fit, the analysis of the correlations becomes challenging.

In addition, it seems very strange that taking all wavelengths or the other set of wavelengths has such an impact. How do you justify this? Is there any physical or instrumental reason to do so?

We do not think it is an instrumental artifact. This is rather caused by fitting three main parameters (ozone, temperature, Ring spectrum) with similar spectral structures, which can compensate one another. The justification of the wavelength sample selected is rather empirical. Our investigations have shown that the retrieval with the preferred wavelengths is least sensitive to the temperature as discussed in Appendix A2 of the revised manuscript.

It would be beneficial to present the product issues with the original WFDOAS algorithm and the impact of all individual changes and/or choices made.

We added an appendix section (Appendix A1) to the revised manuscript presenting details on the algorithm evolution from WFDOAS to WFFA and illustrating the impact of individual changes. We also demonstrate that the retrieval using the selected wavelength sample (odd-numbered) is less dependent on temperature (Appendix A2).

In addition, it is mentioned before that only the central FOV bins, 17 to 20, are used. Can you really assess the across-track variability using only 4 measurements?

As we mentioned above, the total ozone retrieval now covers FOVs from 10 to 22. The across-track variability is now clearly seen.

It is also unclear what is the additional fitted parameter to account for the slope of the ozone absorption signature? Is the polynomial of first order and not a constant? This parameter is not mentioned any longer in the list you give at line 136. Please clarify this.

There are no additional parameters in the WFFA algorithm. Using a cubic polynomial, as in the WFDOAS approach, removes the broad-band spectral signature and retains only the differential absorption structure to be exploited in the fit procedure. The change from the cubic polynomial to a constant allows us to further increase the information gain by including the broad-band absorption structure of ozone. We reworded the manuscript text to avoid confusion.

*O3 profile climatology:*

- While the total column dimension may reliably account for the O3 variability in the stratosphere, this is not the case in the troposphere. There may be significant longitudinal and time variability in the tropospheric ozone content, which is not covered by the climatology as currently built. For example, in Tropics, there is a significant wave-one pattern in the tropospheric columns with an amplitude of up to 20DU. This should be acknowledged and further discussed in the paper. What are the consequences of this limitation on the retrieved total ozone columns?

To investigate this issue, we scaled the lower part of the climatological ozone profiles (below 12 km) by factors 2 and 5 and repeated the retrieval. No significant differences in the resulting total ozone value were identified. This is discussed in Appendix A4 of the revised manuscript.

*Validation:*

- Why do you limit the validation to the period 2016-2018? In section 2, you mention that data from 2012 to 2018 have been processed. It would be beneficial to extend the ground-based validation to the full available period to better evaluate the product stability.

We intend to cover the period from 2012 to 2018. The retrieval is time-consuming; therefore, only the data from 2016 to 2018 are available for now. We modified the expression: "The period for that the ozone data are to be retrieved is intended to cover the years from 2012 until 2018. Currently, only the data from 2016 to 2018 has been retrieved" (now line 66-67).

The added-value of Fig. 8 is limited, especially for differences seen on the orbit edges where sampling differences likely dominate. I encourage making similar plots based on a larger amount of data. The latitudinal dependence aspect is already covered by Figure 9.

The intention of Fig. 8 is to demonstrate how different data sets agree along one sample orbit. The figure allows us to show all four data sets in one plot and demonstrate that the comparison made for the averaged data is consistent with the results for one single orbit.

In Figure 10, the discussion would be easier to follow if you would show similar time series of the relative differences in addition to the current panels. As for the ground-based validation, extending the comparison with the operational OMPS product for the full period 2012-2018 would be beneficial.

The panel of the relative differences that correspond to Figure 10 will be included in the revised manuscript as Figure 11. As explained above, only data for the years 2016 and 2018 are available so far.

**Minor comments**
- p. 2 Line 28: Chiou et al. does not include any DOAS algorithm, but relies on the GTO/Direct-fitting CCI (not DOAS-based) algorithm. The latter is not cited at all while applied to most of the sensors mentioned here. Please correct this and add proper references for DOAS algorithms for each of the sensors.

Only those algorithms related to our study have been mentioned. The references have been corrected.

First, we refer to the instruments in lines 25-28 of the revised version:
"The Global Ozone Monitoring Experiment (GOME, 1995-2011) (Burrows et al., 1999), the SCanning Imaging Absorption spectroMeter for Atmospheric CHartographY (SCIAMACHY, 2002-2012) (Bovensmann et al., 1999) and GOME-2 (2006-present) (Munro et al., 2016) also provide TOC products using the differential optical absorption spectroscopy (DOAS) approach (Lerot et al., 2014)."

Then, in lines 38-39 of the revised manuscript, we refer to the WFDOAS applied to those instruments: "The retrieval approach adapts the Weighting Function-DOAS technique (WFDOAS), successfully applied for SCIAMACHY (Bracher et al., 2005), GOME (Weber et al., 2005) and GOME-2 (Weber et al., 2007)."

- p. 2 Line 41: please add a reference for GOME-2.

Done

p. 2 Line 56: replace as "a three-part instrument, namely a nadir mapper (OMPS-NM), a nadir profiler (OMPS-NP) and a limb profiler (OMPSLP), collecting data since January 2012."

Done

p. 3 Line 66: As the limb profiler is not described, it is not clear to the reader what are the implications to have only data "from the central of the three vertical slits". What does it mean in terms of coverage/spatial resolution?

The text has been reworded as follows (now lines 61-65):

" So far, the limb ozone profiles are only retrieved from the central slit of the three vertical slits of OMPS-LP (Arosio et al., 2018), resulting in a horizontal sampling of about 150 km along-track and 3 km across-track (Algorithm Theoretical Basis Document (ATBD) for the Environment Data Record (EDR) Algorithm of the Ozone Mapping and Profiler Suite (OMPS) Limb Profiler. https://ozoneaq.gsfc.nasa.gov/media/docs/EDR_ATBD_baseline_version1.pdf). In order to match our nadir TOC product to OMPS limb profiles for obtaining tropospheric ozone columns, only the central OMPS-NM across-track FOV bins, 10 to 22, are needed and were processed (approximately 50 km x 600 km wide swath)."

p. 5 Lines 110-115: Please better specify what temperature parameter you fit exactly since I do not think you fit a T° profile but most likely a single parameter.

The text has been changed as follows:

" In Eq. (1) the index i references the wavelengths, $V^t$ is the true vertical ozone column, and $b^t$ are true atmospheric conditions (pressure, temperature, albedo, etc.). $\bar{V}$ is the reference (i.e. used in the forward model) ozone column, $\bar{T}$ is the reference temperature profile and $\bar{b}$ is the atmospheric state as used in the forward model. $\Delta V$ and $\Delta T$ represent the corrections to the reference values which result from the fit. The scalar correction to the temperature profile ($\Delta T$ ) is a shift applied to the entire vertical temperature profile"
(now lines 115-119)

p. 8 Line 192 : 3.5 km x 5.5 km since August 2019.

Done

p. 8 Line 192: Unclear sentence. OFFL and RPRO are produced similarly and both include a cloud correction. Heue et al., 2016 is not an appropriate reference for LIDORT I believe.

The text reads now as follows:

"The L2 product of S5P/TROPOMI used in this study is the offline (OFFL and RPRO) total ozone column product (Lerot et al., 2020). S5P/TROPOMI OFFL and RPRO total ozone are very similar and

are obtained using the GODFIT version 4 retrieval (Lerot et al., 2014). The algorithm performs a direct comparison with simulated radiances through non-linear least-square inversion, using the sun-normalized measured radiance from 325 to 335 nm. The modelled radiances and Jacobians are obtained with the RTM LIDORT (Spurr et al., 2018)." (now lines 203-207).

p. 10 Line 234: "De Bilt" instead of "Debilt »

Since we extended the range of the retrieved FOVs, the comparison with ground-based measurements has changed. This station is not mentioned anymore as the provided maximum negative difference value originates now from another station.

Original sentence:
"The mean relative differences vary from -1.6 % for Debilt (Brewer; 52.1° N, 5.18° E) to 6.0 % for Mauna Loa (Brewer; 19.53° N, 155.57° W)."

New sentence (now L. 243-244):
"The mean relative differences vary from -2 % for Rio Gallegos (Brewer; 51.60° S, 69.32° W) to 4.8 % for Mauna Loa (Brewer; 19.53° N, 155.57° W)."

---

## Author Comment (AC2)

**Atmos. Meas. Tech. Discuss., referee comment RC2**
**Comment on amt-2021-61**
**Anonymous Referee #3**

We appreciate your revision and comments.

After investigations of the spatial patterns of the limb-nadir matching for retrieving tropospheric ozone, we extended the total ozone retrieval to FOVs from 10 to 22. Therefore, the following figures have changed: Figs. 5-7, 9, and 10. The added FOVs do not change main conclusions in the manuscript.

All your technical comments have been addressed and will be included in the revised manuscript. With respect to the major comments, we detail the following points:

**Review comment on**

The paper describes a total ozone Ozone retrieval from OMPS-NM data using a modified DOAS approach.

As preparational work a new ozone climatology has been generated. Which might be interesting in itself, is this available for other user? Has it been compared to existing data sets?

The climatology will be available for download in the same way as the previous one developed by us (Lamsal et al. 2007). We include this information in the revised manuscript in the data availability section. This climatology has not been compared with other datasets.

The algorithm is based on the Weighting Function Differential Optical Absorption Spectroscopy algorithm (WFDOAS), that is adapted to the OMPS-NM. However OMPS-nm has a spectral resolution of 1 nm and a spectral sampling of 0.42 nm, hence the slit function is represented by ~2.2 measurements points, for a classical DOAS analysis this might cause an undersampling issue. The authors solve this issue by skipping half of the spectral points (using only the odd spectral channel) - which gives reasonable results compared to other observations, however no real explanation is given why the approach is working. Moreover, when the other half of the data is used the comparison shows stronger deviation.

The reason to select the wavelength sample is rather empirical. Our investigations have shown that the retrieval with the preferred wavelengths (odd-numbered) is less sensitive to the temperature than any other wavelength combination we tried. We discuss this issue in detail in Appendix A2 of the revised manuscript.

The comparison showed good agreement with operational OMPS and TROPOMI data sets, as well as with ground based Brewer and Dobson measurements.

**general comment**

The analysis is applied to roughly 44 data points (316-336nm) where only half of the data is used. The algorithm is described to become unstable if the complete data range is used. Is its possible that the major deviation are caused by just a few data points? To check this

possibility I suggest to run the analysis for one orbit skipping one even data point after the other. A combination might also be possible but this might easily end up in larger study.

Many sensitivity tests have been carried out to select the optimum fitting window and the wavelength sample. In short, we could not identify any particular spectral point responsible for the observed behaviour. For the final window selected, skipping/adding points at the window's boundaries does not produce significant differences. The use of all the spectral points makes the retrieval much more sensitive to the temperature and typically results in a bias of about -1% to -2%. In the revised version of the paper, we illustrate in detail that the selected wavelength sample results in less dependence on the temperature, which we believe is the main reason for the reduced bias (see Appendix A2)

**minor comments:**

5.3 S5P/TROPOMI

- L 197 This reference is about a tropospheric ozone retrieval but in this context it seems to be a reference on the RTM LIDORT.
The paragraph reads now as follows:

"The L2 product of S5P/TROPOMI used in this study is the offline (OFFL and RPRO) total ozone column product (Lerot et al., 2020). S5P/TROPOMI OFFL and RPRO total ozone are very similar and are obtained using the GODFIT version 4 retrieval (Lerot et al., 2014). The algorithm performs a direct comparison with simulated radiances through non-linear least- square inversion, using the sun-normalized measured radiance from 325 to 335 nm. The modelled radiances and Jacobians are obtained with the RTM LIDORT (Spurr et al., 2018)."  (now lines 203-207).

L 207 why gridding data from two algorithms applied to TROPOPMI spectra for the comparison? Both resulting VCDs have identical coverage. So a direct mapping seems easier.

Due to the large amount of TROPOMI data, it is much more time consuming to make direct comparisons.

6 Validation

L 218 For the comparison of OMPS with TROPOMI the data are again gridded, this probably can not be avoided. But I suggest to use only one gridded TROPOMI data set here.

There is nothing about two TROPOMI datasets in line 218. You probably mean Sect. 6.2. The reason to use only one TROPOMI dataset is, however, unclear from your comment. We prefer to keep both.

Figure 5: The seasonal map shows a strong orbital pattern, which seems surprising when 4 years of data were averaged.

This is because we originally used only four central FOVs considered sufficient for the limb-nadir matching. However, as we mentioned above, the revised version of the manuscript shows FOVs from 10 to 22.  This higher sampling strongly reduced the striping.

6.2 Comparison with OMPS-NM operational product and S5P/TROPOMI

For the OMPS-NM data set only the central field of view was used in the comparison 150km, while for TROPOMI the complete swath was taken into account ~2600 km. I suppose the comparison will improve if also for TROPOMI only the central pixels (~210 to 240) are used.

This issue is reduced by using more FOVs in the OMPS retrieval.

**technical comments**
- Figure 2: [VMR] stands for volume mixing ratio and is hence not a correct unit, please change to [ppm]

Done

- Figure 4: in Figure 3 a positive bias between S5P/TROPOMI WFDOAS relative to the ground based observations is shown, here (in figure4) it seems there is a negative bias of the Operational OFFL data relative to the WFDOAS. I suggest showing WFDOAS - OFFL instead, to have more consistent figures.

We corrected the plot as suggested by the reviewer.

- Figure4: The minus sign at (-10) has disappeared from scale.

Fixed

---

## Author Comment (AC3)

**Comment on amt-2021-61**
**Anonymous Referee #1**

**General Comments**

The authors have implemented a total ozone algorithm for the Ozone Mapping and Profiler Suite (OMPS) Nadir Mapper (NM) based on a technique they call the Weighting Function Fitting Approach (WFFA). The purpose of the algorithm is to estimate total ozone from OMPS NM coincident with OMPS LP stratospheric column ozone, and compute the tropospheric column by taking their difference. As such, the algorithm has been applied only near nadir and in cloud free conditions. The authors explain the WFDOAS approach is not the optimal to retrieve total ozone (TO) from OMPS NM due to the relatively low spectral resolution of the instrument which negatively impacts analysis of the differential spectrum. In the WFFA method the primary algorithmic changes increase the width of the fitting window and reduce the low order polynomial to a constant term. With these modifications, the WFFA method fits the spectral slope of the ozone absorption in addition to higher order structure. The first half of the paper describes the algorithm. The second half details validation of the total ozone retrievals using other datasets that are well known in the field. The total ozone results presented in the second half of the paper look quite good and therefore I feel this algorithm is very promising. However I have some questions about the algorithm and how it has been presented. I recommend the authors address the following to significantly strengthen the paper.

We appreciate your revision and comments.

After investigations of the spatial patterns of the limb-nadir matching for retrieving tropospheric ozone, we extended the total ozone retrieval to cover FOVs ranging from 10 to 22. Therefore, the following figures have changed: Figs. 5-7, 9 and 10. The added FOVs do not change the main conclusions in the manuscript.

All your technical comments have been addressed and will be included in the revised manuscript. With respect to the major comments, we detail the following points:

**Specific Comments**

1. The authors have made an unusual accommodation to get good results from their OMPS NM retrievals - they have achieved their results using only alternating pixels in the OMPS NM spectra. Oddly, this works when odd-numbered pixels are used in the retrieval. When even-numbered or all pixels are used, the results are unstable from one retrieval to the next and a bias is observed. I think it is important in this paper to provide further information on this instability. The issue raises questions about the performance of the algorithm, the instrument, or both. Do the authors have an idea why the WFFA method gives reasonable results in the one specific case? Perhaps the fitting of the spectral slope to determine TO is affected by end-point sensitivity of the fit? The authors use a fitting window of 316-336 nm in their algorithm. Does an adjustment of this window to include/exclude 1-2 spectral points at the window edges produce a more stable retrieval when all spectral pixels are used? Are there any particular spectral features at the edges of the fitting window that complicate a reliable spectral slope determination that might

show as a noticeable pattern in fitting residuals?

The nature of the problem using WFDOAS on OMPS-NM data is the instability of the retrieval resulting in unrealistic behavior of the results across the instrument FOV. We illustrate this effect in the appendix of the revised manuscript.  The WFFA algorithm avoids this obstacle by reducing the weight of the differential absorption structure of ozone in the retrieval and by increasing the weight of the broad-band spectral signature of ozone. This is done by extending the spectral range to 316 to 336 nm and subtracting a lower order polynomial (constant) instead of the cubic one in WFDOAS.  As a result, most of the instabilities has been eliminated, but some of them still remained. Subsequently, we further analyzed the retrieval results obtained using only selected spectral points from the retrieval spectral window.   The spectral sample with every second spectral point was found to have much weaker dependence on the temperature, which made the WFFA retrieval more stable. New plots providing more details on this topic are added to the revised manuscript (Figure A2). We could not identify any particular spectral point or range responsible for the observed behaviour. For the finally selected spectral window and sampling, skipping/adding points at the window's boundaries does not produce any significant differences.

If the authors think the issue is related to
quality of OMPS NM spectra, this should be stated. It is worth nothing that colleagues at
BIRA have successfully retrieved total ozone from OMPS NM with the GODIFT v4 algorithm
to produce data consistent with the GTO-ECV record. I am aware of no similar issues with
processing OMPS NM data.

We do not think it is an instrumental issue. The issue is rather related to the correlation between the spectral signatures of the main fitting parameters, namely, weighting functions of ozone and temperature as well as the Ring spectrum. GODFIT also had issues using the smaller fitting windows. In their final data product, the FOV dependent striping effects have been corrected a-posteriori (C. Lerot, personal communication).

2. The explanation of the insensitivity of the WFFA algorithm to absorbing aerosols and
other broadband contributions should be explained better. The WFFA approach fits the
spectral slope to estimate the ozone absorption signal, but several other geophysical
effects may also affect spectral slope. The authors assume an aerosol-free atmosphere in
their forward model and retrieve an effective scene albedo at 377 nm using the LER
approach, so albedo wavelength is 40 - 60 nm from the edges of the fitting window
region. Absorbing aerosols can produce several percent in spectral dependence
in the radiance signal in this spectral region. The authors state the aerosol effect is largely
accounted by the effective scene albedo, but I feel given the nature of the algorithm this
may be an oversimplication. How can we be better assured of this? It is true that results
shown later there are no significant ozone anomalies in regions of high aerosol load. But I
cannot explain why. The WFFA algorithm may well be as insensitive as authors claim, but
it would be useful for the reader to know the reason(s), and clarify circumstances where
residual error may grow to be significant. Absorbing aerosols are common in the tropical
regions and these are regions where tropospheric ozone is of particular interest. Since
tropospheric column is a relatively small fraction of the total column, small errors for TO
can be non-negligible for tropospheric ozone determination.

In Coldewey-Egbers et al. (2003), "WF-DOAS Algorithm Theoretical Basis Document" (DOI: 10.26092/elib/381), it is shown that: "the effective albedo by the Lambertian Equivalent Reflectivity

(LER) approach near 377 nm represents a first-order correction for non-absorbing aerosols (...) total ozone might be underestimated by 1% if visibility is reduced to 2 km by absorbing aerosols". We repeated this analysis for WFFA algorithm for different boundary layer aerosol types assuming a strong aerosol load (visibility of 2 km) and in addition for an extreme volcanic aerosol load in the stratosphere. We found that the WFFA TOC retrieval errors are highly dependent on the solar zenith angle. For small SZAs (about 30 deg), the TOC might be overestimated by about 3 %, in a presence of weakly absorbing aerosols in the boundary layer. For strongly absorbing (urban) boundary layer aerosols, an overestimation of TOC by about 1 % is found. In the case of an extreme volcanic loading in the stratosphere, the overestimation might reach about 8 %. For high SZAs (about 60 deg), the error is below 0.5 % for weakly absorbing boundary layer aerosols and increasing to about 1%  for strongly absorbing boundary layer aerosols and extreme volcanic aerosol loading in the stratosphere.
The details of this analysis are presented in the Appendix A3 of the revised manuscript.

3. The sensitivity of the algorithm to tropospheric ozone is not discussed in the paper. This should be addressed in some fashion given the goal of the algorithm.

To investigate this issue, we scaled the lower part of the climatological ozone profiles (below 12 km) by factors 2 and 5 and repeated the retrieval. No significant differences in the resulting total ozone value were identified. This is discussed in Appendix A4 of the revised manuscript.

4. Some discussion of algorithm uncertainty and sources of error would strengthen the paper considerably.

A full analysis of uncertainty and errors of WFDOAS was presented in Coldewey-Egbers et al. (2003), "WF-DOAS Algorithm Theoretical Basis Document" (DOI: 10.26092/elib/381). In addition, we re-evaluated the major sources of errors that could be specific for the WFFA retrieval. As a result, we include in the main text of the revised manuscript a table with uncertainty estimates from enhanced aerosol loading, the use of BDM (Malicet) vs Serdyuchenko cross-sections, and tropospheric ozone profile shape (Table 1).

5. It is unclear why S5P/TROPOMI results from different satellite algorithms are compared. How do these comparisons relate to the OMPS-NM WFFA TO algorithm in the present manuscript?

As the WFDOAS algorithm was the basis of WFFA it was worthwhile to include its results for TROPOMI in the comparisons. On the other hand, OFFL/RPRO is the official TROPOMI product and we could not ignore it.

6. The title is very general. A more specific title will help readers distinguish this work from that of others.

The title has been changed.
The new title reads: "Total ozone column from OMPS-NM measurements using the broadband Weighting Function Fitting Approach (WFFA)"

**Techincal comments:**
Line 2: its -> the
Done
14: delete "characterizes the stratosphere. In turn,"

Done
15: remove "On the other hand,"
Done
19: remove "Among others,"
Done
20: "1970's, have provided"
Done
23: specify Suomi NPP OMPS
Done
22: change 1994 to 2005
Done
24: named -> known
Done
24: (all) -> (all instruments)
Done
29: giving -> which is useful to establish
Done
32: this -> that
Done
57: sensor (no s)
Done
58: radiation instead of radiances?
Done
64: 150 km wide swath
Done
95: "linearly"
Done
101: remove comma
Done
Eqn. 1: is this C or C_i?
It is  C. Changed
113, 137: same comment as for Eqn. 1
Done
118: please revise statement in light specific comments above
A detailed discussion of the algorithm evolution from WFDOAS to WFFA and reasons for selecting the wavelength sample are included in the revised version of the manuscript, see Appendix A1
127,128: readouts -> pixels:
In the revised manuscript we avoid using the term "readouts", talking about spectral points instead. (Now lines 128-133)
140: define RTM:
It is defined in L138 of the revised manuscript.
149: this first sentence seems out of place; can safely remove.
We do not understand why this sentence is out of place. It defines the initial guess of total ozone. This value determines the initial guess ozone profile used for the radiative transfer calculations. (Now line 153)
174: may be V8.6:
The OMPS Nadir Mapper level 2 Description cited. It indicates that the version is V8.5
203: "from the" -> "reported with"
Done

212: Is this the IGACO3 recommendation?

No, it is not an IGACO3 recommendation. The line is changed as follows:

Original line: "The S5P-WFDOAS product is retrieved using the recommended Serdyuchenko et al. (2014) cross-sections".

Now line 222: "The S5P-WFDOAS product is retrieved using the Serdyuchenko et al. (2014) ozone absorption cross-sections"

221: Fig. 5 shows ozone lower over Antarctica than tropics during SON.

The line is changed as follows:

Original line: The total ozone reaches its minimum in the tropical region in all seasons increasing polewards.

Now line 231-232: "The total ozone generally shows a minimum in the tropical region in all seasons."

250: "OMPS-L2" does not indicate a specific product. Please clarify which product.

The product version has been specified: OMPS-NM L2 v2.1

284-286: cloud contaminated scenes would generally have low bias, not high

We believe that the reviewer means the bias in the total ozone. The statement refers, however, to the differences between WFFA and OMPS_L2 algorithms. The algorithms might react very differently on the residual cloud contamination. It is, however, impossible to predict if the difference is expected to be positive or negative.

291: define TOCS

Have been changed for TOC along the text.

305: th -> the

Done

327-328: more should be said to justify this statement. What are requirements for retrieving tropospheric ozone from the limb-nadir matching technique?

The lines have been deleted

Fig. 5: striping in these TO maps seems large for a 150 km wide swath.

With only cloud-free pixels processed and limited to four FOVs there are not enough data to remove the striping. The new data set includes 12 FOVs (instead of 4 before) covering approximately 600 km across-track. The striping is now strongly reduced.

Fig. 10: cannot find a reference to this figure in the text. Please define what the shaded areas represent. What is the difference between the grey and the very light green shaded areas?

The shadings indicate the standard deviation of each time series. In the revised manuscript, this is mentioned in the figure caption. The standard deviation of the operational product of OMPS-NM is light green, which turns to grey-green when it overlaps with the WFFA standard deviation shown in grey.

* Minor editing note: the use of plural nouns is not needed in a number of places

All that we could find have been changed.

---

## Author Response (AR2)

**Technical corrections are listed below.**

- Line 6: "operational" instead of "scientific"
  Done

- Line 28: Lerot et al., 2014 is not a good reference for TO3 DOAS retrievals. Instead you can refer for example to Hao et al., AMT, 2014; Van Roozendael et al., JGR, 2006.
  Done

- Line 63: Rault et al. The year of the reference is missing.
  Done

- Line 183: OMPS-NM instead of OMPS-NP
  Done